# The Muscleblind-like protein MBL-1 regulates microRNA expression in *Caenorhabditis elegans* through an evolutionarily conserved autoregulatory mechanism

**Jens Verbeeren** [ID], **Joana Teixeira** [ID], **Susana M. D. A. Garcia** [ID]*

Institute of Biotechnology, HiLIFE, University of Helsinki, Helsinki, Finland

* susanamaria.garcia@helsinki.fi

## Abstract

The Muscleblind-like (MBNL) family is a highly conserved set of RNA-binding proteins (RBPs) that regulate RNA metabolism during the differentiation of various animal tissues. Functional insufficiency of MBNL affects muscle and central nervous system development, and contributes to the myotonic dystrophies (DM), a set of incurable multisystemic disorders. Studies on the regulation of *MBNL* genes are essential to provide insight into the gene regulatory networks controlled by MBNL proteins and to understand how dysregulation within these networks causes disease. In this study, we demonstrate the evolutionary conservation of an autoregulatory mechanism that governs the function of MBNL proteins by generating two distinct protein isoform types through alternative splicing. Our aim was to further our understanding of the regulatory principles that underlie this conserved feedback loop in a whole-organismal context, and to address the biological significance of the respective isoforms. Using an alternative splicing reporter, our studies show that, during development of the *Caenorhabditis elegans* central nervous system, the orthologous *mbl-1* gene shifts production from long protein isoforms that localize to the nucleus to short isoforms that also localize to the cytoplasm. Using isoform-specific CRISPR/Cas9-generated strains, we showed that expression of short MBL-1 protein isoforms is required for healthy neuromuscular function and neurodevelopment, while expression of long MBL-1 protein isoforms is dispensable, emphasizing a key role for cytoplasmic functionalities of the MBL-1 protein. Furthermore, RNA-seq and lifespan analyses indicated that short MBL-1 isoforms are crucial regulators of miRNA expression and, in consequence, required for normal lifespan. In conclusion, this study provides support for the disruption of cytoplasmic RNA metabolism as a contributor in myotonic dystrophy and paves the way for further exploration of miRNA regulation through MBNL proteins during development and in disease models.

## Author summary

Alternative splicing is a molecular process by which one gene can produce multiple protein isoforms. These isoforms often have different properties, and normal cell function

---

**Data Availability Statement:** All relevant data are available from within the manuscript as well as the supplemental information file. The Gene

Expression Omnibus (GEO) reference series number for the RNA-seq and RIP-seq data originating from this study is GSE249410 and includes accession numbers GSE249406, GSE249407, and GSE249408.

**Funding:** This work and S.M.D.A.G. were supported by the Academy of Finland Project Funding (grant number 309173), University of Helsinki Internal Funding. J.V. was supported by an Ella and Georg Ehrnrooth Foundation Grant (https://www.ellageorg.fi). J.T. was funded by the Instrumentarium Science Foundation (https://instrufoundation.fi) (230040) and the Finnish Cultural Foundation (https://skr.fi) (00222470). The funders had no role in study design, data collection and analysis, decision to publish, or preparation of the manuscript.

**Competing interests:** The authors have declared that no competing interests exist.

and development requires expression of the appropriate isoform. The regulation of alternative splicing is carried out by a set of specialized proteins, one of which is the Muscleblind-like protein (MBNL). This protein is essential during the development and differentiation of many animal tissues. We found that this protein regulates the alternative splicing of its own gene in animals as divergent as nematode and human. We showed that this leads to the timely expression of MBNL protein isoforms that localize to the cytoplasm of the cell. In the cytoplasm, these isoforms acquire new functions unrelated to splicing and regulate molecular processes that control the expression of many other genes. These functions are required for normal development of the nervous system and lifespan in nematodes. This finding is particularly important because it helps us understand the regulation and function of MBNL proteins in animal development as well as identify potential disease mechanisms in myotonic dystrophies, neurodegenerative disorders associated with MBNL dysfunction.

## Introduction

Precise and timely regulation of gene expression is essential for normal organismal development and response to environmental changes. This regulation occurs in part at the post-transcriptional level through the action of RNA-binding proteins (RBPs), which recognize specific sequences in target RNAs. RBPs modify transcripts through various processes such as alternative splicing and polyadenylation, and regulate their transport, translation and stability [1]. RBPs are also involved in the biogenesis of microRNAs (miRNAs) [2], a class of small non-coding RNAs that negatively regulate gene expression through binding to complementary sites in target mRNAs. RBPs play key functions in the regulation of gene expression essential for neuronal development and synaptic plasticity [3, 4], as well as control biological programs that drive stem cell renewal and differentiation [5]. As a result, disruption of RBPs is often associated with neurodegeneration and cancer [3, 6]. Thus, comprehensive studies of the regulatory mechanisms that govern RBP function are crucial to a better understanding of their role in organismal development and in mechanisms of pathogenesis.

The Muscleblind-like (MBNL) protein family is a group of ancient metazoan RBPs [7], required for various reprogramming pathways in vertebrates during post-natal heart development [8], and for stem cell [9], erythroid [10] and myofibroblast [11] differentiation. They are characterized by the presence of one or two evolutionarily conserved tandem CCCH-type zinc finger (TZF) motifs [12, 13], which confer recognition to 5′-YGCY-3′ binding motifs in target RNAs [14]. Originally identified as regulators of alternative splicing [15, 16], MBNL proteins have also important functions in the regulation of alternative polyadenylation [17, 18], mRNA transport [19–21] and stability [22, 23]. Humans, like most vertebrates, have three Muscleblind-like genes (Muscleblind-Like Splicing Regulator 1, 2 and 3; *MBNL1*, *2* and *3*) [13, 24], encoding for proteins mainly expressed in skeletal muscle and nervous system [15, 24, 25]. Functional disruption of MBNL proteins is associated with a number of disease features in the neuromuscular degenerative disorders myotonic dystrophy type 1 and 2 (DM1 and DM2) [26], where MBNL proteins are sequestered in nuclear foci by RNAs bearing expanded CUG or CCUG repeats, respectively [24, 25].

Given the importance of RBPs in post-transcriptional regulation, their gene expression is under strict control. RBPs often bind their own RNA transcripts, resulting in alternatively spliced mRNA isoforms with reduced stability [27]. This autoregulatory mechanism prevents deleterious changes in RBP expression levels. Vertebrate *MBNL* genes, however, employ a

unique type of autoregulation in which alternative splicing of the *MBNL* transcript generates protein isoforms with distinct subcellular localizations [8, 28–30]. Specifically, MBNL binding inhibits inclusion of an exon within its own transcript, thereby disrupting a bipartite nuclear localization signal that consists of a repetition of consecutive lysine-arginine amino acid residues (referred to as KR motif) [31]. As the level of MBNL rises during development, a progressive shift occurs from longer protein isoforms, located exclusively in the nucleus, to shorter isoforms with a more cytoplasmic distribution [8–10, 28, 29, 31–35]. While a study has identified cytoplasmic MBNL as a promoter of neurite outgrowth [36], a focus on the functional significance of nuclear versus cytoplasmic MBNL isoform expression *in vivo* and in a whole-organismal context is lacking.

In the nematode *Caenorhabditis elegans*, the single *MBNL* orthologous gene *mbl-1* encodes a protein with a prominent neuronal expression [37–39], where it is required for synapse formation in DA9 motor neurons [39] and dendritic morphogenesis in the PVD sensory neuron [40]. MBL-1 has been shown to regulate alternative splicing [41, 42], but its role in other processing pathways has not been demonstrated. Here, we show conservation of autoregulatory alternative splicing in the *C. elegans mbl-1* gene, revealing it as a key regulatory feature of the *MBNL* gene family. We further suggest a model in which MBNL protein isoforms have evolutionarily conserved functional roles. Our aim was, taking advantage of this evolutionary conservation and the genetic amenability of *C. elegans*, to shed light on the regulatory principles that govern expression of MBNL proteins in a whole-organismal context, and to uncover the biological functions of the isoforms. During neuronal development, a transition occurs from longer to shorter MBL-1 isoforms, and we identify the Rbfox-family ortholog FOX-1 as a tissue-specific regulator of *mbl-1* alternative splicing. The MBL-1 isoforms exhibit distinct subcellular localizations and functions, with the longer isoforms localizing solely to the nucleus and shorter isoforms expressed in both the nucleus and cytoplasm. Moreover, we show that the expression of shorter MBL-1 isoforms is required for normal neuromuscular function, neurite outgrowth, and, through the regulation of a subset of miRNAs, for normal lifespan. Together, our data uncover an expanded regulatory role for the shorter, cytoplasmically expressed MBL-1 isoforms with implications for human tissue-specific gene regulation and disease.

## Results

### 1. Autoregulatory control of Muscleblind-like protein isoform localization is conserved in *C. elegans*

In the *C. elegans mbl-1* gene, alternative splicing and alternative promoter usage generate several distinct mRNA isoforms [38]. Our RT-PCR analysis of *mbl-1* transcripts detected eight mRNA isoforms from a mixed-stage population using forward primers targeting two distinct promoter regions (Fig 1A). For isoforms 1 to 4, usage of upstream promoter 2 and activation of a start codon located in exon 1 leads to the addition of 56 amino acids to the N-terminal tail. Exon sequences, the surrounding splice site signals, and the encoded amino acids (AAs) show significant conservation in the nematode lineage (S1 Fig). Alternative splicing at the 3′ end of the gene results in mRNA isoforms that not only differ in the coding sequence for the last nine C-terminal AA residues, but also in the length and composition of their 3′-UTRs. Splice site signals, and sequences in exons and the 3′-UTR again show considerable conservation in nematodes (S2 Fig). Finally, alternative splicing regulates the inclusion of exon 7 (encoding an extra 75 AAs) to generate either long(ex7+) or short(ex7-) MBL-1 protein isoforms. Interestingly, we detected the presence of a KR motif encoded by exon 7 in proximity to a second KR motif encoded by exon 8. In human MBNL proteins, a double KR motif constitutes a nuclear localization signal (NLS) [31]. Phylogenetic analysis revealed that sequences encoding a KR motif

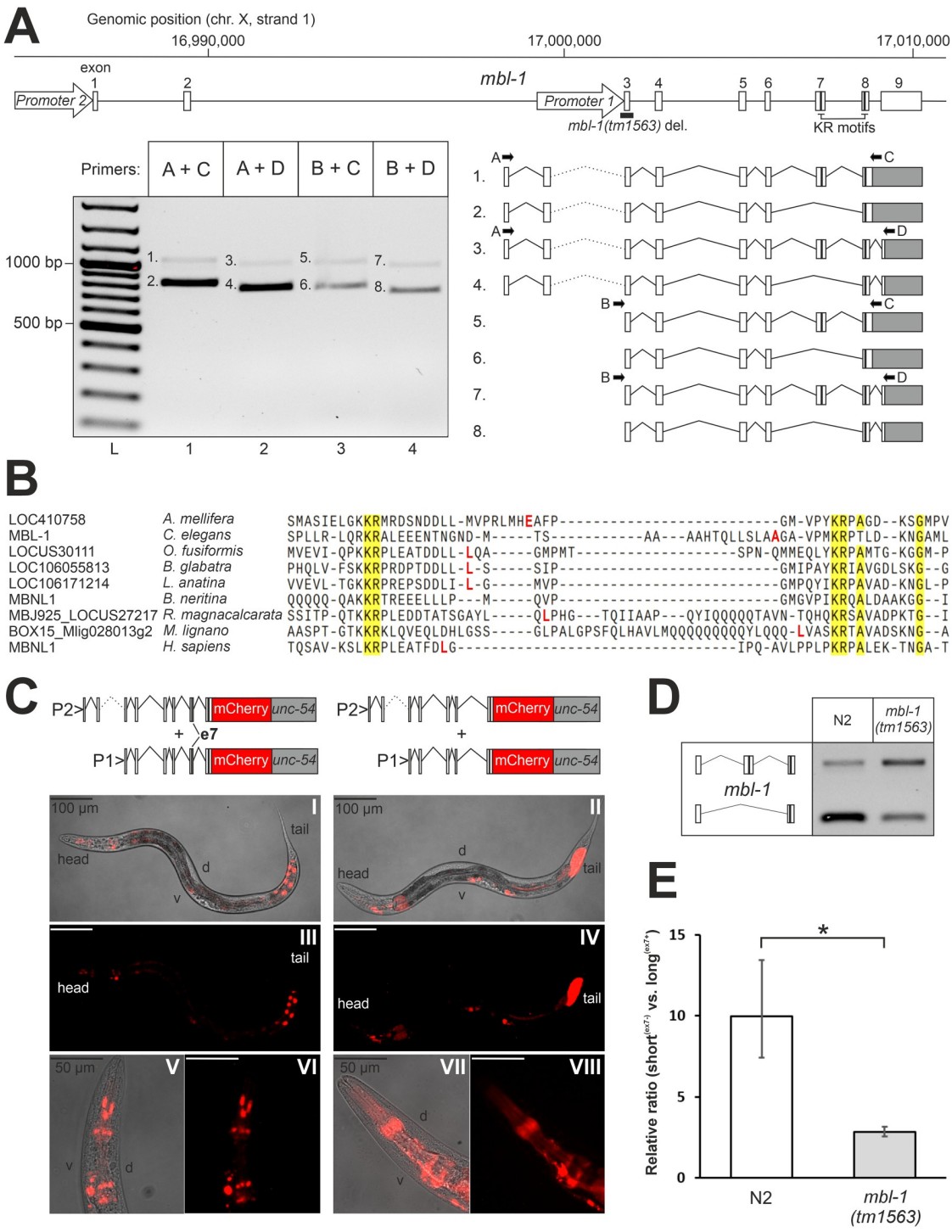

**Fig 1. Autoregulation of *mbl-1* generates protein isoforms with distinct subcellular localizations.** (A) Depiction of the *mbl-1* genomic region and splicing variants detected by PCR at the L4 stage. Arrows indicate the location of promoter regions 1 and 2 reported in [39]. Thick black line indicates deleted region in the *mbl-1(tm1563)* mutant. Black vertical lines indicate locations of the KR motifs, and the grey area indicates the 3′-UTR. "L" indicates DNA ladder. Numbered bands were sequenced and correspond to numbered isoforms. (B) Multiple sequence alignment of KR motif-encoding exons from orthologous *MBNL* genes in representative species from different bilaterian phyla. Amino acid sequence conservation > 85% is highlighted in yellow, and the amino acid residue upstream exon-exon junction is indicated in red (no data for *B. neritina*). (C) Representative differential interference contrast (DIC) and fluorescence images of animals at the L4 stage expressing mCherry tagged with either long$^{(ex7+)}$ or short$^{(ex7-)}$ MBL-1 isoforms. Fusion constructs were expressed under both promoters 1 and 2 (P1 and P2) together, and with the *unc-54* 3′-UTR

(in grey). "D" and "v" indicate dorsal and ventral sides, respectively. (D) RT-PCR analysis of *mbl-1* exon 7 alternative splicing in wild-type (N2) and *mbl-1(tm1563)* animals at the L4 stage. (E) qPCR quantification of the ratio of exon 7 skipping versus inclusion. Experiments were done with three biological repeats (each with three technical replicates) with RNA collected from animals grown on different plates. An unpaired two-sample Student's *t*-test was performed (* $P < 0.05$). Error bars indicate standard error of the mean (SEM).

repeat are present and located on adjacent exons in orthologous *MBNL* genes within all phyla of the bilaterian clade, except for the Ambulacraria (Fig 1B and S3 Fig). In addition, we found that the exon encoding the first KR motif is consistently subject to alternative splicing. Evolutionary conservation of exon sequences across bilaterians is almost exclusively limited to the KR motifs (Fig 1B). In contrast, the MBNL protein family-specific tandem CCCH-type zinc finger motifs 1 and 2 (TZF1 and TZF2) show a higher degree of conservation predating the emergence of the Bilateria (S4 Fig).

We asked whether inclusion of exon 7 into the *mbl-1* transcript generates protein isoforms that localize to the nucleus, analogous to the nuclear localization mechanism for human MBNL proteins. To address this, we generated strains that express either long$^{(ex7+)}$ or short$^{(ex7-)}$ MBL-1 protein isoforms, tagged with mCherry, under both endogenous promoters together, and investigated their subcellular localization at the L4 larval stage (when most tissue development is complete). Expression of MBL-1 was detected in various tissues in a pattern consistent with previous studies [38, 39], specifically in the head and tail ganglia, the ventral nerve cord (VNC), the spermatheca, and the posterior gut (Fig 1C: panels I to IV). Consistent with the human longer MBNL isoform, mCherry signals showed a distinct nuclear localization when linked with long$^{(ex7+)}$ isoforms, most noticeably in, but not limited to, cells of the gut and the head region (Fig 1C: panels I, III, V and VI). In contrast, short$^{(ex7-)}$ isoforms showed a more diffuse pattern, consistent with localization both in the nucleus and cytoplasm (Fig 1C: panels II, IV, VII and VIII). We note that the expression in the gut might be ectopic and a possible consequence of the presence of the *unc-54* 3′-UTR, as reported before [43].

In human MBNL proteins, MBNL regulates the alternative splicing of its own transcript to generate shorter protein isoforms with a disrupted NLS [29]. To investigate whether alternative splicing of exon 7 is autoregulatory, we employed a loss-of-function *mbl-1*(*tm1563*) mutant strain [39], with a 512 base pair (bp) deletion eliminating the 121 bp long exon 3 (Fig 1A), and analyzed exon 7 alternative splicing at the L4 stage through qPCR. Absence of functional MBL-1 protein consistently promoted production of long$^{(ex7+)}$ isoforms (Fig 1D and 1E).

Collectively, our data confirm the existence of eight distinct *mbl-1* mRNA isoforms. Inclusion of exon 7 encoding an evolutionarily conserved KR motif into the *C. elegans mbl-1* transcript leads to the nuclear localization of the resulting MBL-1 protein isoform, analogous to the nuclear localization mechanism for human MBNL proteins. Moreover, MBL-1 itself inhibits inclusion of exon 7 and thereby promotes short$^{(ex7-)}$ isoforms with a more cytoplasmic distribution.

## 2. Alternative splicing of the *mbl-1* transcript exhibits tissue-specific regulation during larval development

We next aimed to examine how the expression of long$^{(ex7+)}$ and short$^{(ex7-)}$ MBL-1 isoforms is developmentally regulated, and whether regulation occurs in a tissue-specific manner. To address these questions, we set out to visualize alternative splicing of *C. elegans mbl-1* exon 7 *in vivo*. We generated a bichromatic fluorescence alternative splicing reporter strain that expresses two reporter constructs under the control of the endogenous *mbl-1* promoter 1 (Fig

2A). We tested expression under promoter 2 and observed similar tissue expression, although expression under promoter 1 occurred in a wider range of tissues (S5 Fig). The splicing reporter allowed us to distinguish between exon 7 inclusion and skipping, generating respectively a green fluorescent protein (GFP) or a red mCherry signal. We observed tissue-specific variations in mCherry-GFP fluorescence intensity ratios during development, supporting the presence of regulatory tissue specificity (Fig 2B). Exon skipping (red) was predominant in the excretory canal cell at all developmental stages. Exon inclusion (green) was preferred in the posterior gut at all stages, as well as during vulval development at the L3 and L4 stages. Switching of fluorophore genes between the constructs showed a concordant change in signal ratios (S6 Fig), and expression in the gut persisted after swapping the *unc-54* 3′-UTR of the reporter genes with the *let-848* 3′-UTR (S7 Fig).

Motor neurons in the VNC displayed a shift from predominantly exon inclusion (ca. 60% GFP) at the L1 stage to mostly exon skipping (ca. 56% mCherry) at the L4 stage (Fig 2C and 2D). Conversely, the excretory canal cell strongly favored exon skipping (ca. 90% mCherry) throughout all larval stages (Fig 2C and 2E). We then asked whether additional factors regulate alternative splicing of *mbl-1* exon 7 in the excretory canal cell. To identify potential regulatory elements near exon 7, we looked for conserved binding motifs within the exon and its flanking introns through multiple sequence alignment with five different *Caenorhabditis* species (S8 Fig). We identified a conserved 5′-GCAUG-3′ sequence element in the upstream intron, corresponding to a potential binding site for FOX-1, the *C. elegans* ortholog of the mammalian RNA-binding Rbfox proteins [44]. Strikingly, crossing of a loss-of-function *fox-1(y303)* mutant [45] into our alternative splicing reporter strain reduced exon skipping in the excretory canal cell to ca. 73% (Fig 2F and 2G). The fluorescent color change in *fox-1(y303)* animals was specific to the excretory canal cell, as no such decrease occurred in the VNC (Fig 2F and S9 Fig).

Taken together, our alternative splicing reporter showed that, during development, *mbl-1* alternative splicing displays tissue-specific regulation. In the VNC, alternative splicing of the *mbl-1* transcript exhibits a developmental shift towards short$^{(ex7-)}$ MBL-1 isoforms. In the excretory canal cell, FOX-1 co-regulates alternative splicing of the *mbl-1* transcript inhibiting inclusion of exon 7.

## 3. Short$^{(ex7-)}$ MBL-1 expression is required for normal lifespan, neuromuscular function, and neurite outgrowth

To investigate the functional relevance of *mbl-1* exon 7 inclusion, we employed targeted CRISPR/Cas9 genomic editing and generated two strains that express exclusively either long$^{(ex7+)}$ isoforms (*mbl-1(syb4345)*, referred to as *mbl-1 long$^{(ex7+)}$*), or short$^{(ex7-)}$ isoforms (*mbl-1(syb4318)*, referred to as *mbl-1 short$^{(ex7-)}$*). Isoform-specific expression at the *mbl-1* locus was confirmed by RT-PCR (Fig 3A), and *mbl-1* transcript expression levels were similar for all strains (S10 Fig). Next, we measured the relevance of exon 7 inclusion to organismal fitness indicators such as lifespan, motility, and neuronal development.

Previous research has shown that the absence of functional MBL-1 reduces lifespan [37, 46]. Strikingly, normal lifespan was restored only when short$^{(ex7-)}$ MBL-1 isoforms were expressed (mean lifespan wild-type N2 = 21.5 days vs. *mbl-1 short$^{(ex7-)}$* = 21.6 days) (Fig 3B). In contrast, long$^{(ex7+)}$ isoform expression barely (6%) restored lifespan (mean lifespan *mbl-1 (tm1563)* = 14.5 days vs. *mbl-1 long$^{(ex7+)}$* = 15.4 days). Next, in our assessment of organismal fitness, we tested motility by performing a thrashing (swimming) assay. L4 stage *mbl-1 (tm1563)* animals showed a clear impaired thrashing phenotype, indicative of defective neuromuscular function. Remarkably, and similarly to the lifespan assay, exclusive expression of

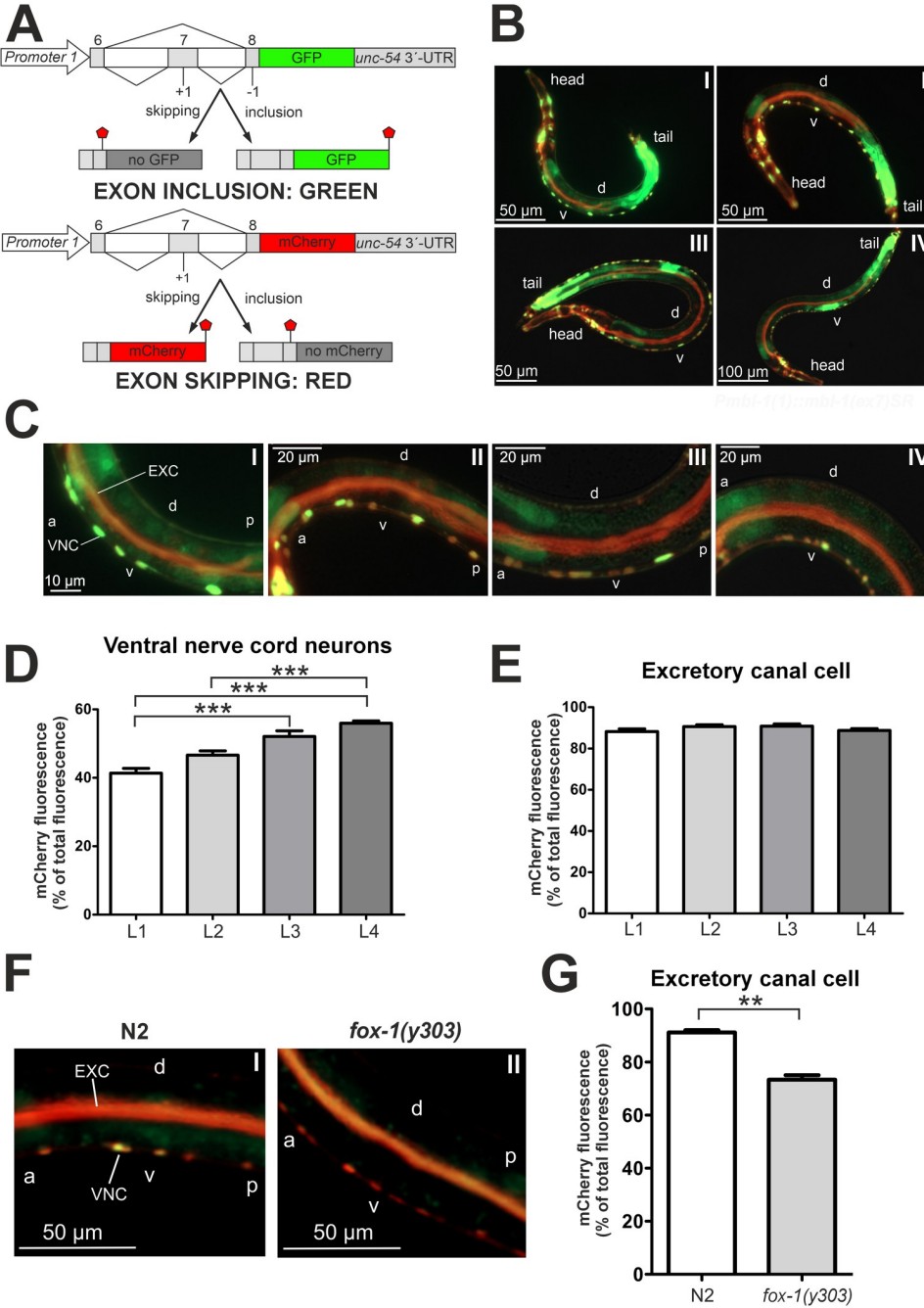

**Fig 2. Alternative splicing of the *mbl-1* transcript is developmentally regulated and co-regulated by FOX-1.** (A) Schematics show the gene constructs used to generate the alternative splicing reporter strain. A genomic region including exon 6, 7 and 8 of the *mbl-1* gene was cloned in frame with genes expressing GFP and mCherry, respectively. Stop signs indicate location of stop codons. (B) Representative fluorescence images of the alternative splicing reporter strain taken at the L1 (panel I), L2 (II), L3 (III) and L4 (IV) stages. "D" and "v" indicate dorsal and ventral sides, respectively. (C) Fluorescence images of animals expressing the alternative splicing reporter minigenes during L1 (panel I), L2 (II), L3 (III) and L4 (IV) stages. VNC indicates ventral nerve cord and EXC excretory canal cell. "A" indicates anterior, "p" posterior, "d" dorsal and "v" ventral side of the animal. (D) Quantification of relative mCherry fluorescence intensity in VNC neurons of L1 (n = 23), L2 (n = 25), L3 (n = 22) and L4 (n = 25) stage animals. A Kruskal-Wallis test with Dunn´s multiple comparison test was performed (*** $P < 0.001$). Error bars indicate SEM. (E) Quantification of relative mCherry fluorescence intensity in the excretory canal cell of L1 (n = 23), L2 (n = 25), L3 (n = 22) and L4 (n = 25) stage animals. One-way ANOVA with Tukey´s multiple comparison test was performed. No significance was found. Error bars indicate SEM. (F) Fluorescence images of N2 and *fox-1(y303)* animals at the L4

stage expressing the fluorescence alternative splicing reporter minigenes. VNC indicates ventral nerve cord and EXC excretory canal cell. "A" indicates anterior, "p" posterior, "d" dorsal and "v" ventral side of the animal. (G) Quantification of relative mCherry fluorescence intensity in the excretory canal cell from animals in (F). For each strain, a total population of 30 worms was analyzed, grown on two different plates (15 animals per plate). An unpaired two-sample Student's *t*-test was performed (** $P < 0.01$). Error bars indicate SEM.

short(ex7-) MBL-1 isoforms resulted in a full rescue of the motility phenotype (Fig 3C). Interestingly, animals expressing only long(ex7+) MBL-1 isoforms displayed an intermediate phenotype.

Effects on neuronal development were assessed through the characterization of the gentle touch responsiveness. L4 stage *mbl-1(tm1563)* animals showed a significant reduction of the gentle touch responsiveness (Fig 3D), suggesting dysfunction of the touch receptor neurons (TRNs) [47], whereas both *mbl-1* isoform-specific mutants showed no defects. To visualize subtle defects in the TRN network, we crossed the *mbl-1* mutant strains into a reporter strain that expresses GFP in the TRNs, *zdIs5*[*Pmec-4*::*GFP*]. At the L4 stage, *mbl-1(tm1563)* animals displayed a robust reduction in posterior lateral microtubule cell (PLM) outgrowth, resulting in an increase of the receptive field gap between the anterior lateral microtubule cell (ALM) and PLM neurons (Fig 3E and 3F). Expression of short(ex7-) MBL-1 isoforms restored the gap to wild-type level, whereas expression of long(ex7+) MBL-1 isoforms showed a mild defect.

Collectively, our data show that expression of the short(ex7-) MBL-1 isoforms is required for normal *C. elegans* lifespan, neuromuscular function, and neuronal development, whereas the expression of long(ex7+) MBL-1 isoforms is dispensable.

## 4. Long(ex7+) MBL-1 isoforms have a higher splicing regulatory activity than short(ex7-) isoforms

To identify the molecular mechanisms that could underlie the observed phenotypic differences, we performed RNA-seq of wild-type (wt) N2 animals and all *mbl-1* mutant strains at the L4 stage. Since MBNL proteins are known regulators of alternative splicing [16, 20], we performed an initial splicing analysis on the RNA-seq data. This would enable us to identify targets of MBL-1 alternative splicing regulation, and to evaluate the differential impact of the MBL-1 isoforms on alternative splicing. We identified 235 splicing events that were differentially regulated in *mbl-1(tm1563)* (adjusted $P < 0.01$), relative to wt (Fig 4A and S1 Table). These included 168 cases of exon skipping, 36 cases of mutual exclusive exon selection, and 19 and 12 cases of alternative 5′ splice site and 3′ splice site selection, respectively. In *mbl-1 short(ex7-)* animals, 84 of the 148 (56.8%) differentially spliced events were shared with *mbl-1(tm1563)* mutants. In *mbl-1 long(ex7+)* animals, we found 89 differentially spliced events, of which 29 (34.5%) were shared with *mbl-1(tm1563)* mutants. Tissue enrichment analysis showed that most dysregulated splicing events in *mbl-1(tm1563)* animals occur in genes predominantly expressed in neuronal cells, and GO enrichment analysis indicated that target genes are most often associated with actin-binding processes and the adherens junction (Fig 4B).

To compare the splicing regulatory activities of the MBL-1 isoforms that either include or skip exon 7, we defined an isoform-specific splicing rescue index (SRI) for a given splicing event as one minus the ratio of two ΔPSIs (change in percentage spliced in, relative to wt):

$$SRI(iso) = 1 - \frac{\Delta PSI(iso)}{\Delta PSI(mut)}$$

Here, ΔPSI(iso) relates to the isoform-specific (short(ex7-) or long(ex7+)) ΔPSI for a specific splicing event, and ΔPSI(mut) to the ΔPSI of mutant *mbl-1(tm1563)* animals for the same

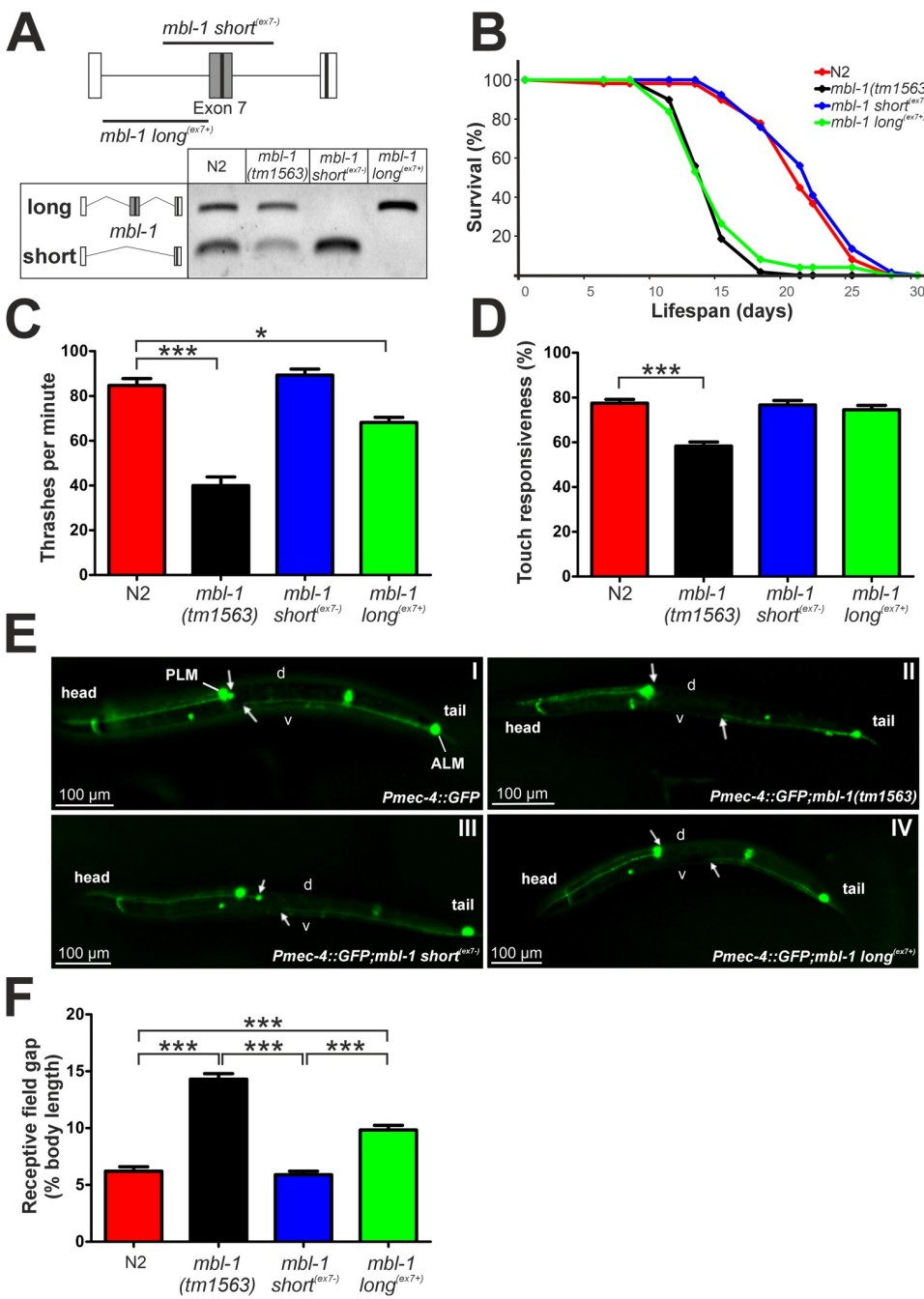

**Fig 3. Short$^{(ex7-)}$ MBL-1 isoform expression is required for normal lifespan, thrashing and neurite outgrowth processes in *C. elegans*.** (A) Schematic depiction of CRISPR/Cas9-targeted deletion at the *mbl-1* genomic locus to generate strains that express either only short$^{(ex7-)}$ or only long$^{(ex7+)}$ MBL-1 isoform, and RT-PCR verification of single isoform expression in these strains. Black boxes indicate locations of the KR motifs. (B) Lifespan curves of N2 and *mbl-1* mutant animals fed with HT115 bacteria at 20°C. Both *mbl-1(tm1563)* and *mbl-1 long$^{(ex7+)}$* animals have shortened lifespan compared to N2 and *mbl-1 short$^{(ex7-)}$* animals ($P < 0.00001$, log-rank test). (C) Results from the thrashing assay of L4 stage N2, *mbl-1(tm1563)*, *mbl-1 short$^{(ex7-)}$*, and *mbl-1 long$^{(ex7+)}$* animals derived from three biological repeats with animals grown on three different plates (total animal population: N2 = 108, *mbl-1(tm1563)* = 93, *mbl-1 short$^{(ex7-)}$* = 165, mbl-1 long$^{(ex7+)}$ = 99). One-way ANOVA with Dunnett´s multiple comparison test was performed (* $P < 0.05$, *** $P < 0.001$). Error bars indicate SEM. (D) Results from the gentle touch response assay of N2, *mbl-1 (tm1563)*, *mbl-1 short$^{(ex7-)}$*, and *mbl-1 long$^{(ex7+)}$* L4 stage animals. For each strain, 60 animals grown on three different plates were analyzed. One-way ANOVA with Dunnett´s multiple comparison test was performed (*** $P < 0.001$). Error bars indicate SEM. (E) Representative fluorescence images of strains at the L4 stage expressing GFP in the touch

receptor neurons (TRNs) crossed with indicated *mbl-1* mutant strains. White arrows indicate the location of the ALM cell body and neurite end of the PLM. The gap in between is the receptive field gap. "D" and "v" indicate dorsal and ventral sides, respectively. (F) Measurement results of the receptive field gap length from L4 stage *mbl-1* mutant animals crossed with the TRN GFP reporter strain. For N2, *mbl-1(tm1563)*, *mbl-1 short*(ex7-), and *mbl-1 long*(ex7+) strains respectively 57, 69, 63 and 66 animals grown on three different plates were analyzed. A Kruskal-Wallis test with Dunn´s multiple comparison test was performed (*** $P < 0.001$). Error bars indicate SEM.

splicing event. An SRI of "one" thus constitutes complete splicing rescue. Strains expressing the short(ex7-) and long(ex7+) MBL-1 isoforms displayed on average SRIs of 0.57 and 0.89, respectively (Fig 4C and S2 Table). RT-PCR analysis of three representative splicing events for which *mbl-1 short*(ex7-) animals showed either weak (*blmp-1*; SRI < 0.33), intermediate (*unc-43*; 0.33 < SRI < 0.66), or high (*unc-36*; SRI > 0.66) rescue in the RNA-seq analysis, validated these results (Fig 4D).

To evaluate the contribution of the different MBL-1 isoforms to *mbl-1* autoregulation, we assessed for changes in mCherry and GFP expression of our alternative splicing reporter strain crossed with the mutant *mbl-1* strains. Exon 7 skipping was highly dependent on the presence of long(ex7+) MBL-1 isoforms in the VNC, demonstrated by the predominantly green fluorescence in animals that lack this isoform, *mbl-1(tm1563)* and *mbl-1 short*(ex7-) (Fig 4E and 4F). In contrast, we did not observe a similar dependence on long(ex7+) isoforms in the excretory canal cell (S11 Fig).

In summary, splicing analysis of our RNA-seq data identified 235 splicing events dependent on the presence of MBL-1. Through our SRI analysis, we showed that the splicing regulatory activity of long(ex7+) MBL-1 isoforms is higher than that of short(ex7-) isoforms. In addition, our results established that long(ex7+) MBL-1 isoforms are the principal isoforms to carry out autoregulatory alternative splicing.

## 5. MBL-1 binding stabilizes microtubule-associated transcripts

Next, to identify specific molecular pathways regulated by MBL-1 and its isoforms, we carried out gene expression analysis of the RNA-seq data. This revealed 797, 1670 and 860 differentially expressed genes in *mbl-1(tm1563)*, *mbl-1 long*(ex7+) and *mbl-1 short*(ex7-) animals (false discovery rate (FDR) < 0.01, > 1.5-fold change), respectively (S3 Table). Interestingly, *mbl-1 (tm1563)* and *mbl-1 long*(ex7+) animals share 466 differentially expressed genes, constituting 58.5% of all differentially expressed genes in *mbl-1(tm1563)* mutants (Fig 5A). In contrast, only 125 (15.7%) differentially expressed genes were shared between *mbl-1(tm1563)* and *mbl-1 short*(ex7-) animals.

We then subjected up- and downregulated genes to gene ontology (GO) analysis with the gene enrichment analysis tool on the WormBase website [48, 49]. Among upregulated genes in *mbl-1(tm1563)* and *mbl-1 long*(ex7+) animals, the most significantly enriched were genes involved in nucleosome, molting cycle and protein hetero-dimerization (Fig 5B). Surprisingly, given their apparent phenotypic similarity to the wild-type strain, *mbl-1 short*(ex7-) animals still show a significant number of dysregulated genes. The vast majority of these were upregulated (759 up- vs. 101 downregulated), and were mostly associated with meiotic cell cycle, reproductive processes, and phosphorus metabolic processes. Using the WormExp database [50], built from published high-throughput expression data, we compared the transcriptional profile of the *mbl-1 short*(ex7-) strain with mutants in the database and found that it was most similar to that of a *set-9;set-26* mutant [51], defective in H3K4me3 histone modification (S4 Table). Interestingly, in *mbl-1 long*(ex7+) animals, downregulated genes are often involved in rRNA metabolism and biogenesis, whereas in *mbl-1 short*(ex7-) animals, many downregulated genes

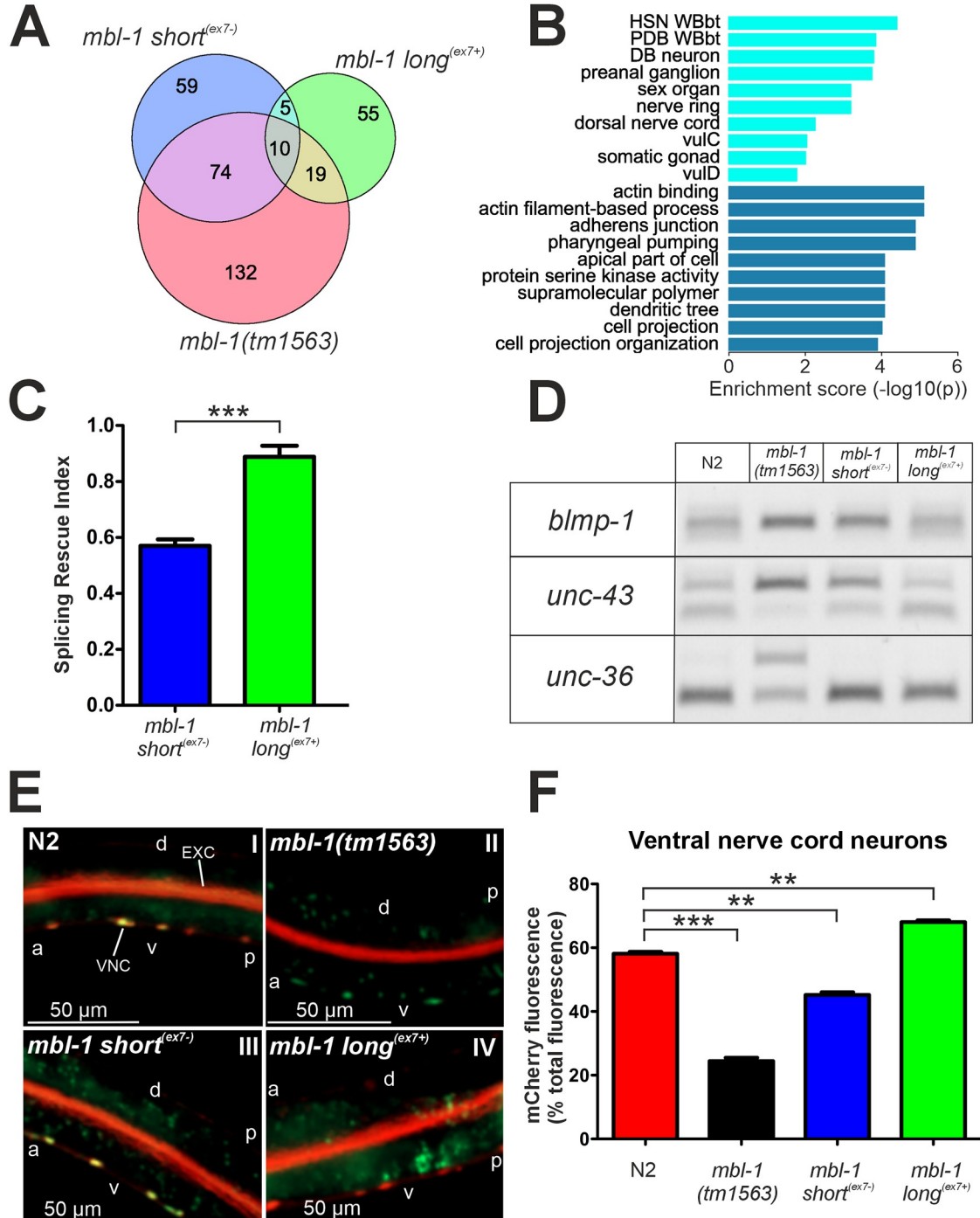

**Fig 4. Splicing regulatory activity of long^(ex7+) MBL-1 isoforms is higher than that of the short^(ex7-) isoforms.** (A) Venn diagram showing overlap in differentially regulated splicing events between *mbl-1* mutants compared to N2 at the L4 stage. (B) Tissue (light blue) and gene ontology (dark blue) enrichment analysis of genes with dysregulated splicing events in *mbl-1(tm1563)* animals. (C) Splicing rescue index (SRI) values of *mbl-1 long^(ex7+)* and *mbl-1 short^(ex7-)* animals for splicing events dysregulated in *mbl-1(tm1563)* animals. The SRI for *mbl-1 long^(ex7+)* animals was significantly higher than that of *mbl-1 short^(ex7-)* animals (0.89 vs. 0.57, *** $P < 0.001$, unpaired two-sample Student's *t*-test). (D) PCR validation of splicing events in *blmp-1* ($SRI_{short(ex7-)} = 0.26$, $SRI_{long(ex7+)} = 0.71$), *unc-43* ($SRI_{short(ex7-)} = 0.36$, $SRI_{long(ex7+)} = 1.01$), and *unc-36* ($SRI_{short(ex7-)} = 0.87$, $SRI_{long(ex7+)} = 1.04$) genes for indicated strains at L4 stage. (E) Fluorescence images of N2 and *mbl-1* mutant animals at the L4 stage expressing the fluorescence alternative splicing reporter minigenes. VNC indicates ventral nerve cord and EXC excretory canal cell. "A" indicates anterior, "p" posterior, "d" dorsal and "v" ventral side of the animal. (F) Quantification of relative mCherry fluorescence intensity in the VNC from animals

in (E). For each strain, a total population of 30 worms was analyzed, grown on two different plates (15 animals per plate). A Kruskal-Wallis test with Dunn's multiple comparison test was performed (** $P < 0.01$; *** $P < 0.001$). Error bars indicate SEM.

are involved in the immune response. Strikingly, among the most significantly downregulated genes in *mbl-1(tm1563)* mutants were *ben-1*, *mec-7*, *mec-12*, and *mec-17*, coding for structural components and regulators of the microtubule cytoskeleton (Fig 5C). These results were validated by qPCR, except for *ben-1* (Fig 5D). Moreover, these genes were also downregulated in *mbl-1 long*(ex7+) animals, suggesting that expression of short(ex7-) MBL-1 isoforms is sufficient for their normal expression. However, qPCR analysis did not conclusively corroborate this.

To reveal direct binding targets of MBL-1, we employed RNA Immunoprecipitation Sequencing (RIP-seq). For this, we generated CRISPR/Cas9-edited animals with a 3xFLAG peptide sequence, followed by a mCherry coding sequence inserted in frame with the MBL-1 coding region, *mbl-1c(syb5299)* (Fig 6A). The modified MBL-1 showed wild-type transcript levels (S10 Fig) and efficient splicing, as splicing defects characteristic of an *mbl-1* knockout strain [41] were not observed in our 3xFLAG::MBL-1 strain (S12 Fig). In addition, mCherry expression was consistent with MBL-1 expression patterns described in previous reports (S13 Fig) [38, 39]. Western Blot following RNA immunoprecipitation at the L4 stage showed specific binding to the anti-FLAG magnetic beads (Fig 6B). To identify interactions with the whole RNA population, immunoprecipitated and input RNA were subjected to RNA-seq without poly(A) enrichment step. We identified 127 transcripts derived from protein coding genes that are bound by MBL-1 (S5 Table). Tissue enrichment analysis revealed that pulled down transcripts represented genes mostly expressed in the nervous system, particularly in VNC motor neurons (Fig 6C). Top enriched GO terms were supramolecular polymer, cell body, and alternative mRNA splicing. Transcripts represent many microtubule-associated genes (*ben-1*, *mec-7*, *mec-12*, *mec-17*, *tba-1*, *tbb-2*), as well as V-type proton ATPase subunit genes (*vha-2*, *vha-3*, *vha-5*, *vha-8*, *vha-11*, *vha-16*) that are mainly expressed in the excretory canal cell [52] (Table 1). MBL-1 also binds transcripts from other RNA splicing-associated genes, as well as its own mRNA transcript, showing direct biochemical evidence of autoregulation. Of the 127 transcripts bound by MBL-1, 15 transcripts derived from genes differentially expressed in *mbl-1(tm1563)* animals (six up- and nine downregulated), and another 12 transcripts came from genes with dysregulated splicing in the same animals (S5 Table). Therefore, these transcripts represent primary targets of MBL-1, regulating their stability or alternative splicing. Finally, we found transcripts from 64 non-coding RNA genes bound by MBL-1, the largest fraction of which were snoRNAs.

In conclusion, our RNA-seq data suggested that most of the genes and pathways affected by the absence of MBL-1 are due to lack of functional MBL-1 short(ex7-) isoforms. Furthermore, we provided evidence that microtubule-associated gene expression is dysfunctional in *mbl-1 (tm1563)* animals. Finally, RIP-seq analysis indicated that transcripts from both tubulin alpha (*tba-1*, *mec-12*) as well as beta (*ben-1*, *mec-7*, *tbb-2*) chain subunit genes are direct targets of MBL-1 regulation, as are transcripts from various V-type proton ATPase subunit genes and RNA-splicing associated genes.

## 6. Short(ex7-) MBL-1 isoforms regulate miRNA expression and consequently are required for normal lifespan

To yield better insights on the functional differences between the MBL-1 isoforms from our RNA-seq gene expression data, we utilized the WormExp database and compared the transcriptional profile of the *mbl-1* mutants to those from mutants present in the database. We found a strongly significant overlap ($P < 1E-10$; Fisher's exact test) between upregulated genes

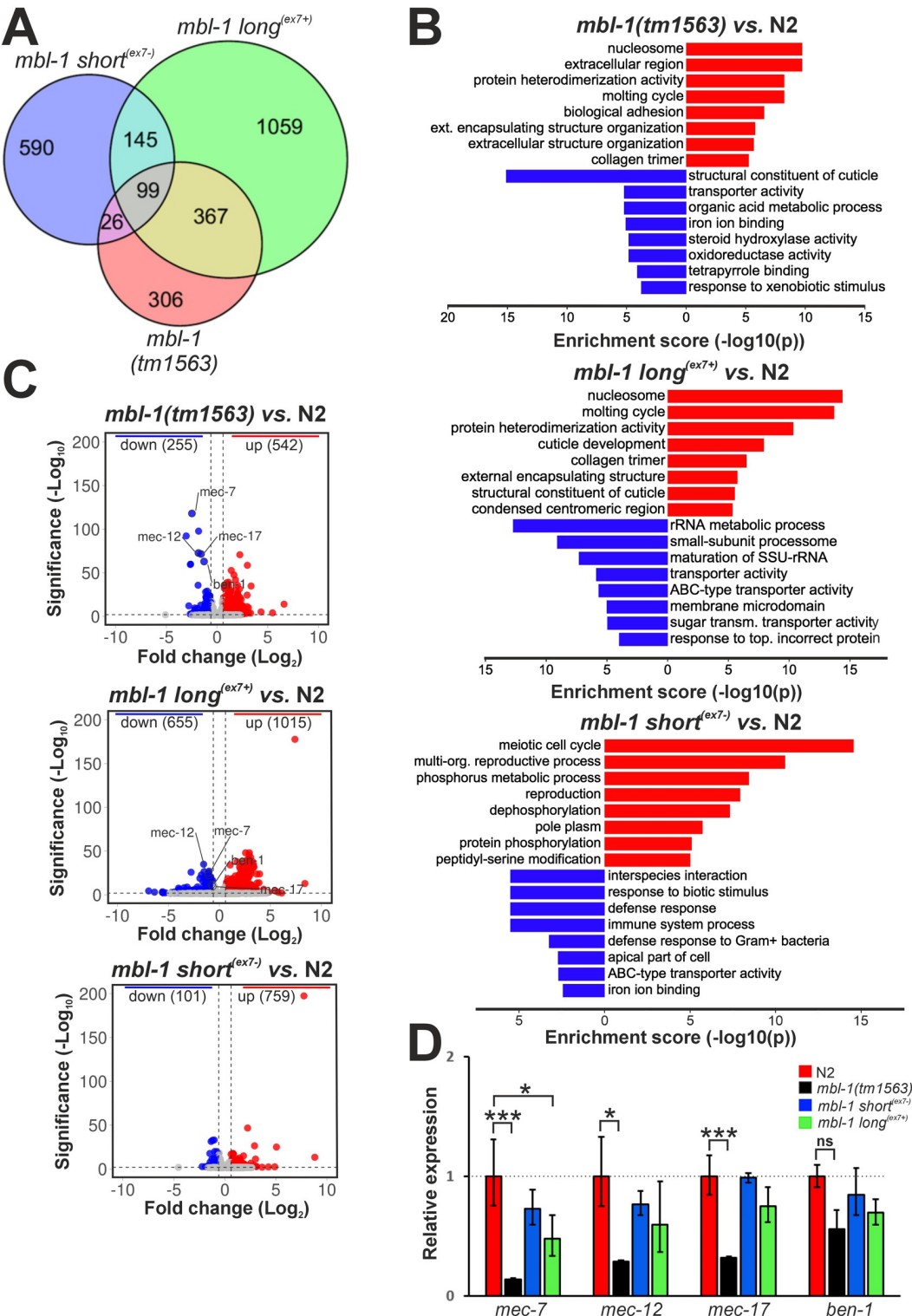

**Fig 5. Microtubule-associated mRNAs are downregulated in the *mbl-1(tm1563)* strain.** (A) Venn diagram showing overlap in differentially expressed genes between *mbl-1* mutants compared to N2 at the L4 stage. (B) GO enrichment analysis for the *mbl-1* mutants compared to N2. (C) Volcano plots showing log₂ fold change (x-axis) and significance (y-axis) of genes differentially expressed in the *mbl-1* mutants. Horizontal dashed line ($>$ 2) indicates significance threshold (FDR $>$ 0.01), vertical dashed lines ($>$ 0.58 and $<$ -0.58) indicate fold change threshold (fold change $>$ 1.5). (D) qPCR validation of selected genes. Experiments were performed with three biological repeats (each with three technical replicates) with RNA collected

from animals grown on different plates. One-way ANOVA with Dunnett´s multiple comparison test was performed (* $P < 0.05$, *** $P < 0.001$, ns: no significance). Error bars indicate SEM.

from a knockout *alg-1(gk214)* mutant [53] and downregulated genes in the *mbl-1(tm1563)* and *mbl-1 long(ex7+)* animals (Fig 7A). In contrast, we did not find a significant overlap with downregulated genes in *mbl-1 short(ex7-)* animals. Argonaute-Like Gene-1 (ALG-1) is the primary mediator of the miRNA pathway [54], and therefore upregulated genes in the *alg-1(gk214)* mutant strain constitute potential direct targets of miRNA regulation. We argued that downregulation of these genes might indicate disruption of miRNA regulation in both *mbl-1(tm1563)* and *mbl-1 long(ex7+)* animals. To test this hypothesis, we performed small RNA-seq analysis of all *mbl-1* mutants at the L4 stage. In *mbl-1(tm1563)* mutants, 39 miRNAs were significantly differentially expressed (> 1.5-fold change, FDR < 0.05), with miRNA expression both up– and downregulated (Fig 7B and S6 Table). In the strain exclusively expressing the long MBL-1 isoform, we found 38 differentially expressed miRNAs. Strikingly, these strains displayed a strong overlap of 16 miRNAs (Fig 7C), 13 of which showed dysregulation in the same direction (Table 2). In the *mbl-1 short(ex7-)* animals, in contrast, we found only 10 differentially regulated miRNAs, with only one miRNA (miR-37-3p) derived from the guide strand of the miRNA duplex and thus functionally loaded with Argonaute proteins. Furthermore, p-values were higher and fold changes markedly lower than for the other *mbl-1* mutant strains (Fig 7B), indicating a far weaker disruption of miRNA regulation in the *mbl-1 short(ex7-)* animals.

In *C. elegans*, a subset of miRNAs are known regulators of lifespan [56]. We hypothesized that the presence of miRNA dysregulation in *mbl-1(tm1563)* and *mbl-1 long(ex7+)* animals contributes to their decrease in lifespan. This would render these animals less susceptible to further lifespan reduction after additional miRNA disruption. To investigate the effect of complete disruption of miRNA regulation on the lifespan of *mbl-1* mutants, we treated animals with *alg-1* RNAi. We noticed a dramatic reduction in lifespan in both wt and *mbl-1 short(ex7-)* animals

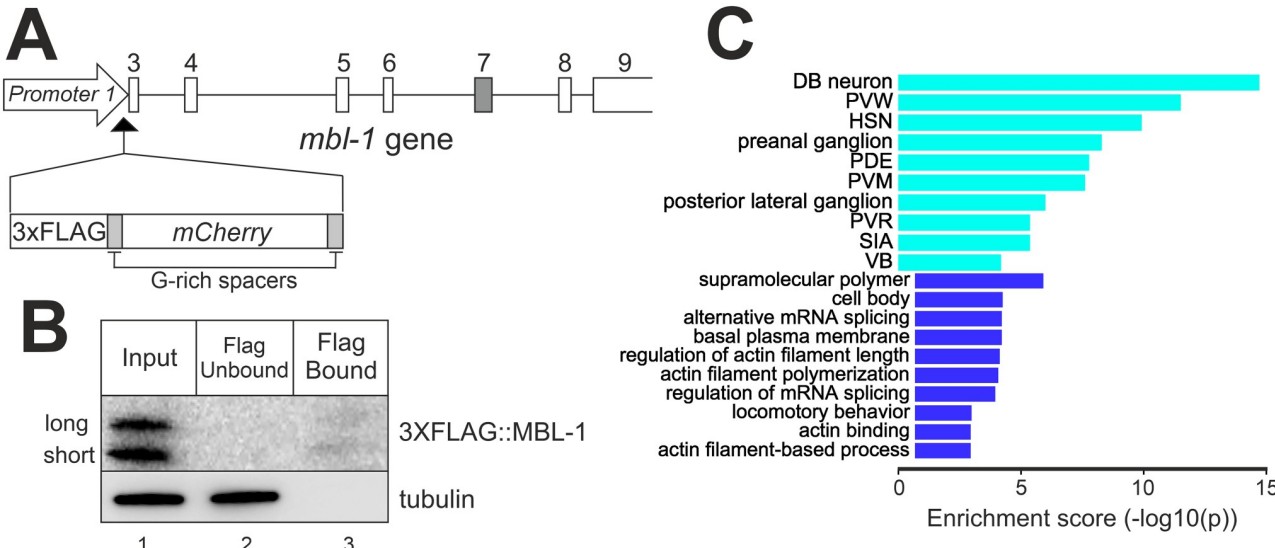

**Fig 6. MBL-1 binds RNA transcripts of neuronally expressed genes.** (A) Schematic indicating CRISPR/CAs9-directed integration of 3xFLAG and mCherry sequences into the *mbl-1* genomic locus. Exon 7 in dark grey. (B) Western blotting analysis of immunoprecipitated MBL-1 from the 3XFLAG-tagged MBL-1 expressing animals at the L4 stage. (C) Tissue (light blue) and gene ontology (dark blue) enrichment analysis of genes whose RNA transcripts were pulled down together with MBL-1.

**Table 1. Summary of genes with transcripts pulled down together with MBL-1.**

| 127 Protein-coding genes | |
| --- | --- |
| **Gene** | **Gene description** |
| Microtubule-associated genes (6) | |
| *ben-1* | Tubulin beta chain |
| *mec-7* | Tubulin beta-1 chain |
| *mec-12* | Detyrosinated tubulin alpha-3 chain |
| *mec-17* | Alpha-tubulin N-acetyltransferase 1 |
| *tba-1* | Tubulin alpha chain |
| *tbb-2* | Tubulin beta-2 chain |
| RNA splicing-associated genes (5) | |
| *asd-2* | KH-domain containing RNA-binding protein |
| *fox-1* | Sex determination protein |
| *mbl-1* | Muscleblind-like protein |
| *rbm-25* | RRM protein-containing protein |
| *sup-12* | PIWI domain-containing protein |
| V-type proton ATPase genes (6) | |
| *vha-2* | V-type proton ATPase 16 kDa proteolipid subunit 3 |
| *vha-3* | V-type proton ATPase 16 kDa proteolipid subunit 2/3 |
| *vha-5* | V-type proton ATPase subunit A |
| *vha-8* | V-type proton ATPase subunit E |
| *vha-11* | V-type proton ATPase subunit C |
| *vha-16* | V-type proton ATPase subunit D |
| **64 Non-coding RNA genes** | |
| snoRNAs (39) | |
| snRNAS (7) | |
| miRNA (1) | |
| other (17) | |

For a complete list, see S5 Table.

with *alg-1* RNAi (Cox proportional hazards regression (HR): HR wt = 42.94, 95% CI 10.38–178; HR *mbl-1 short*$^{(ex7-)}$ = 11.72, 95% CI 6.25–21.97), with the mean lifespan dropping from 23.0 and 21.6 days to 14.9 days in both strains (Fig 7D). This effect was far less pronounced for *mbl-1(tm1563)* and *mbl-1 long*$^{(ex7+)}$ animals (Cox proportional hazards regression: HR *mbl-1 (tm1563)* = 2.80, 95% CI 1.79–4.39; HR *mbl-1 long*$^{(ex7+)}$ = 2.45, 95% CI 1.69–3.57), with a reduction in mean lifespan from 15.2 and 16.0 days to 12.8 and 13.3 days, respectively.

In summary, our small RNA-seq data revealed that strains lacking short$^{(ex7-)}$ MBL-1 isoforms show dysregulation of a significant set of miRNAs. In addition, our lifespan analysis indicated that strains expressing short$^{(ex7-)}$ MBL-1 isoforms are more sensitive to miRNA dysregulation through *alg-1* RNAi knockdown than those that lack these isoforms. Taken together, our data showed that expression of short$^{(ex7-)}$ MBL-1 isoforms is required for optimal regulation of miRNA pathways in *C. elegans* and, as a result, for normal lifespan.

## Discussion

Our results in *C. elegans* demonstrate the evolutionary conservation of an autoregulatory mechanism present in *MBNL* genes that employs alternative splicing to generate two distinct protein isoform types. These protein isoform types exhibit different subcellular localizations and functions. During development, an isoform transition occurs from the initial expression

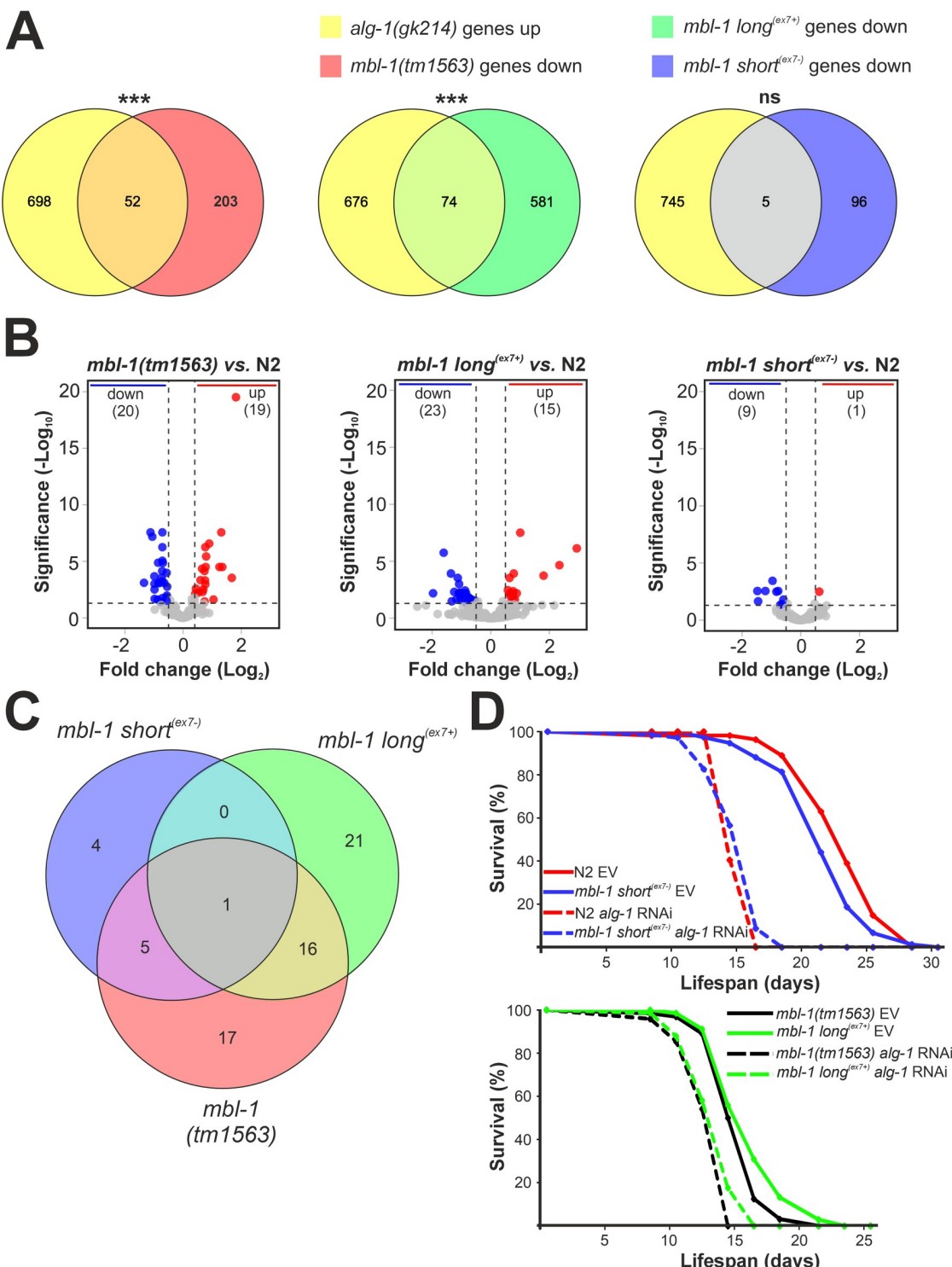

**Fig 7. Short^(ex7-) MBL-1 isoforms regulate miRNAs involved in lifespan pathways.** (A) Venn diagram showing overlap between upregulated genes in *alg-1(gk214)* [53] and downregulated genes in *mbl-1* mutant animals at the L4 stage. Genes from *mbl-1 (tm1563)* and *mbl-1 long^(ex7+)* animals show significant overlap (*** $P$ < 1E-10, Fisher's exact test, ns = no significance). (B) Volcano plots showing log$_2$ fold change (x-axis) and significance (y-axis) of miRNAs differentially expressed in *mbl-1* mutant animals. Horizontal dashed line (> 1.30) indicates significance threshold (FDR > 0.05), vertical dashed lines (> 0.58 and < -0.58) indicate fold change threshold (fold change > 1.5). (C) Venn diagram showing overlap in differentially expressed miRNAs between *mbl-1* mutant animals relative to N2 at the L4 stage. (D) Lifespan curves of N2 and *mbl-1* mutant animals at 20˚C fed with HT115 dsRNA-expressing bacteria either targeting *alg-1* or expressing the control vector L4440 (EV: empty vector).

**Table 2. Dysregulated miRNAs shared between *mbl-1(tm1563)* and *mbl-1 long*[(ex7+)].**

| miRNA | FC (Log$_2$) *mbl-1(1563)* | FC (Log$_2$) *mbl-1 long*[(ex7+)] | Function |
|---|---|---|---|
| Upregulated in both *mbl-1(tm1563)* and *mbl-1 long*[(ex7+)] | | | |
| miR-243-3p | +0.90 | +1.00 | Dauer formation [55] |
| miR-788-5p | +1.35 | +2.93 | Unknown |
| miR-90-3p | +1.31 | +0.58 | Unknown |
| let-7-3p* | +1.04 | +2.34 | Unknown |
| miR-1830-5p* | +0.77 | +0.76 | Unknown |
| miR-230-3p 232323023333ptretrrp3p | +0.79 | +0.71 | Unknown |
| Downregulated in both *mbl-1(tm1563)* and *mbl-1 long*[(ex7+)] | | | |
| miR-246-3p | -0.71 | -1.08 | Lifespan regulation [56] |
| miR-1817 | -0.72 | -1.12 | Unknown |
| miR-359 | -0.72 | -1.06 | Hypoxia resistance [57] |
| miR-40-5p* | -0.85 | -1.02 | Unknown |
| miR-39-5p | -0.99 | -1.09 | Unknown |
| miR-85-3p | -0.82 | -0.95 | Stress response [58] |
| miR-1829b-3p* | -1.35 | -1.17 | Unknown |
| Opposite regulation in *mbl-1(tm1563)* and *mbl-1 long*[(ex7+)] | | | |
| miR-245-3p | +1.82 | -0.88 | Unknown |
| miR-790-5p | +1.25 | -1.24 | Unknown |
| miR-251 | +1.67 | -1.96 | Innate immunity [59] |

* Passenger strand-derived miRNA; FC: fold change.

of long, nuclear MBNL protein isoforms towards shorter, cytoplasmically expressed isoforms. In *C. elegans*, expression of short[(ex7-)] MBL-1 isoforms is necessary for normal lifespan, neuro-muscular function, and outgrowth processes in neuronal tissue, whereas expression of long[(ex7+)] isoforms is not required. In addition, we show that these shorter isoforms also regulate miRNA expression, with targeted miRNAs possibly having roles in lifespan pathways.

Through which mechanism(s) do short[(ex7-)] MBL-1 isoforms regulate miRNA expression? The cytoplasmic localization of the short[(ex7-)] isoforms suggests a role as regulator of cytoplasmic processing steps within the miRNA biogenesis pathway. In the cytoplasm, the RNAse Dicer cleaves the pre-miRNA hairpin to a miRNA duplex and the functional strand is loaded with Argonaute proteins into the RNA-induced silencing complex (RISC) [60]. Short[(ex7-)] MBL-1 isoforms could regulate these processing steps through their affinity for single- and double-stranded RNA 5′-YGCY-3′ motifs [14, 33] present in target pre-miRNA hairpins or miRNA duplexes. In the human heart, MBNL1 binds a single UGC motif within the loop of the pre-miR-1 hairpin, preventing LIN28-mediated degradation of pre-miR-1 [61]. Of note, in animals lacking MBL-1, we found no disruption of expression of miR-1, a miRNA conserved in all bilaterians [62]. As an alternative mechanism, short[(ex7-)] MBL-1 isoforms might regulate miRNAs at the level of miRNA gene transcription. Interestingly, RIP-seq analysis showed that MBL-1 directly binds the *lin-42* transcript (S5 Table). The *lin-42* gene is a homolog of the circadian rhythm gene *period* of Drosophila and mammals, and encodes a transcriptional repressor of a large group of miRNAs that regulate developmental timing [63, 64]. Animals that lack short[(ex7-)] MBL-1 isoforms show a two-fold increase of *lin-42* expression levels (S3 Table). This suggests that these short isoforms could regulate miRNA expression through control of *lin-42* transcript stability, though further studies are required to elucidate this interaction. Whatever

the regulatory mechanism may be, short[(ex7-)] MBL-1 isoform activity causes both a decrease and increase in abundance of select miRNAs (Table 2), similarly to HRPK-1, an ortholog of human hnRNPK and another RBP known to interact with miRNAs in *C. elegans* [65].

Previous research has shown that MBL-1 is required for normal lifespan in *C. elegans* [37, 46]. Our results demonstrate that the presence of short[(ex7-)] MBL-1 is required for normal lifespan (Fig 3B). In addition, lifespan assay analysis after abrogation of miRNA function, taken together with our small RNA-seq data, suggests that short[(ex7-)] MBL-1 isoforms could regulate miRNAs implicated in lifespan pathways. From our set of miRNAs dysregulated in both *mbl-1 (tm1563)* and *mbl-1 long[(ex7+)]* mutants, miR-246-3p is, to our knowledge, the only miRNA previously identified in lifespan regulation. This miRNA is a positive regulator of lifespan [56, 66] and accordingly, is downregulated in these *mbl-1* mutants. It would be of interest to evaluate the contribution of miR-246-3p, as well as those of other dysregulated miRNAs, to lifespan pathways potentially regulated by short[(ex7-)] MBL-1 isoforms.

In addition to their requirement for normal lifespan, the shorter, more cytoplasmically expressed MBL-1 isoforms are also required for neurite outgrowth during the development of the touch receptor neurons (TRNs) in *C. elegans*. However, we could not detect clear *mbl-1* expression in the TRNs with our splicing reporter, and thus we were not able to identify MBL-1 isoform preference in the TRNs and connect isoform expression with the TRN receptive field mutant results. Similar to our results, in cultured hippocampal neurons, expression of cytoplasmic human MBNL1, rather than nuclear, enhances axon outgrowth and dendrite length [36]. Importantly, microtubule-associated gene expression is dysregulated in animals lacking the short[(ex7-)] isoforms (Fig 5C). Specifically, both *mec-7/β-tubulin* and *mec-12/α-tubulin* genes, expressed at high levels in the TRNs [67, 68] and whose loss-of-function alleles cause neurite growth defects [69], are downregulated. MBL-1 directly associates with *mec-7* and *mec-12* transcripts, providing additional evidence for MBL-1 as a regulator of their stability (Table 1), though our approach could not discriminate which MBL-1 isoform type carries out this function. A study showing regulation of *mec-7* and *mec-12* mRNA stability by MBL-1 in touch neurons supports our findings [70]. Furthermore, RIP-seq also showed MBL-1 association with transcripts of alpha-chain tubulin (*tba-1*), beta-chain tubulin (*ben-1*, *tbb-2*) and tubulin acetyltransferase (*mec-17*) genes, thereby suggesting a role for MBL-1 as an important regulator of cytoskeleton dynamics.

Dysregulation of human MBNL proteins plays a role in the pathogenesis of myotonic dystrophy type 1 and 2 (DM1 and DM2) [25, 71–73]. In DM1 myoblasts, all MBNL1 protein isoforms are recruited into characteristic ribonuclear foci by CUG-repeats [32]. The resulting MBNL dysregulation causes splicing errors that correlate with several clinical features [74]. In this context, it is intriguing to note that animals that exclusively express short[(ex7-)] MBL-1 isoforms display improved lifespan, neuromuscular activity and neuronal outgrowth compared to animals that express only long[(ex7+)] MBL-1 isoforms, even though splicing of MBL-1 target transcripts is far more efficient in the latter (Fig 4C). We therefore suggest that the cytoplasmic activity of MBL-1 is crucial for these processes, whereas reduced splicing regulatory activity is not detrimental. Our results in a whole-organismal context provide support for the disruption of cytoplasmic RNA metabolism as a significant contributor to disease in myotonic dystrophy disorders.

In mammals and birds, a developmental transition from longer to shorter MBNL isoforms occurs in various tissues [8–10, 29, 33–35]. As MBNL concentration increases during development, progressive binding to its own mRNA template inhibits inclusion of a 54 bp long exon, driving the isoform shift. Correspondingly, we confirmed conservation of the isoform shift in *C. elegans*, most notably during development of the motor neurons in the VNC from larval stages L1 to L4 (Fig 2C and 2D). Since *mbl-1* alternative splicing disrupts a conserved bipartite

nuclear localization signal [31], the conserved MBL-1 protein isoform switch fundamentally represents a shift from nuclear to cytoplasmic functionalities. In early stages of development, nuclear MBL-1 would serve particularly in the regulation of splicing, although we also found evidence of a role as a regulator of alternative polyadenylation through skipping of an alternative terminal exon (S14 Fig). Evidently, our SRI analysis showed higher splicing regulatory activity for long[(ex7+)] MBL-1 isoforms (Fig 4C), accompanied by a higher autoregulatory splicing activity (Fig 4E and 4F). This likely pertains to the strictly nuclear localization of these isoforms, although additional sequences derived from exon 7 may contribute to the higher splicing regulatory activity. Ultimately, the developmental shift unlocks the cytoplasmic functionalities associated with the short[(ex7-)] MBL-1 isoforms, including the regulation of miRNA expression. Since a single miRNA can bind target sequences in the 3′-UTRs of up to a hundred or more different mRNAs [75], implications on gene expression can be profound. Importantly, even though the inappropriate expression of short[(ex7-)] MBL-1 isoforms in our *mbl-1 short*[(ex7-)] mutant did not cause any obvious phenotypic defects, the expression of 860 genes was dysregulated. The vast majority of these genes (759) were upregulated, and we hypothesize that this occurs either through binding of their transcripts by short[(ex7-)] MBL-1 isoforms or through indirect mechanisms. Of note, our RIP-seq analysis identified three transcripts bound by MBL-1 and upregulated in *mbl-1 short*[(ex7-)] animals (*dao-2*, *osm-11*, and *T19H5.4*: S3 and S5 Tables). Interestingly, upregulated genes in *mbl-1 short*[(ex7-)] animals showed an extremely significant overlap with upregulated genes in mutants defective in H3K4me3 histone modification (*set-9;set-26* and *wdr-5* mutants: S4 Table), enriched for germline specific genes, suggesting potential antagonism of the short[(ex7-)] MBL-1 isoforms with histone modification pathways.

Our PCR analysis identified eight different *mbl-1* mRNA isoforms (Fig 1A). Our work focused on the shared properties and functions of a combination of MBL-1 protein isoforms that either include or exclude exon 7 encoding the conserved KR motif. Additional studies are required to resolve the functional relevance of the individual MBL-1 isoforms expressed from promoter 2 and resulting from alternative splicing at the 3′ end of the *mbl-1* gene, as well as their respective contributions to lifespan, neuromuscular activity, and neurodevelopment. It is interesting to note that alternative promoter regions also exist in *Drosophila* and human *MBNL* genes [76], perhaps reflecting a requirement of MBNL to respond to different regulatory transcription factors, or to allow for regulated tissue-specific expression. Interestingly, alternative splicing at the 3′ end of the *mbl-1* transcript also changes the length and the composition of the 3′-UTR. Intron retention generates mRNA isoforms with a longer 3′-UTR featuring two conserved binding sites for miR-79 and the miR-58/80/81/82 miRNA family, respectively (S2 Fig). Therefore, to gain further insight into *mbl-1* regulation, it would be important to evaluate the distinctive regulatory effects of these miRNAs on *mbl-1* mRNA isoforms that differ in the composition of their 3′-UTRs.

We identified co-regulation of autoregulatory alternative splicing of the *mbl-1* transcript through another RBP, the Rbfox-family ortholog FOX-1 (Fig 2F and 2G). This co-regulation has important effects on the developmental MBL-1 isoform shift in the excretory canal cell, where FOX-1 inhibits the inclusion of exon 7, analogously to MBL-1. In mouse embryonic stem cells, Rbfox2 acts as a crossregulator of negative feedback loops to tune autoregulation of splicing factors [77]. Therefore, our result provides *in vivo* evidence that crossregulation of autoregulatory splicing is a conserved feature of the Rbfox protein family. Furthermore, RIP-seq data revealed direct binding of MBL-1 to *fox-1* transcripts (Table 1). These interactions might indicate reciprocal crossregulation to fine-tune tissue-specific splicing networks. Interestingly, during pluripotent stem cell differentiation in vertebrates, both MBNL1 and RBFOX2 collaborate on an alternative splicing program regulating separate splicing events [9]. Similarly

to *mbl-1* exon 7 (Fig 2B), inclusion of the 54 bp exon in vertebrate *MBNL1* and *MBNL2* transcripts varies significantly in different embryonic tissues [9, 29, 33, 34], suggesting tissue-specific variation of MBNL expression levels or co-regulation through other RBPs. Both *C. elegans* exon 7 and its mammalian counterpart are located within genomic regions of high conservation (S8 Fig) [78]. We anticipate that these sequences harbor binding sites for RBPs that, depending on physiological and cell-specific context, co-regulate alternative splicing of the Muscleblind-like transcript.

Our work gives prominence to the functional versatility of the MBNL protein family and shows evolutionary conservation in *C. elegans* of its known roles as regulators of alternative splicing and polyadenylation, as well as mRNA stability. These various functional properties depend on the subcellular localization of the MBNL protein isoforms, determined, in turn, by a unique autoregulatory mechanism. A phylogenetic analysis demonstrates that the functional requirements for autoregulation (the KR motifs that determine localization and the alternative splicing of associated exons) are present in orthologous *MBNL* genes within virtually all phyla of the bilaterian clade (S3 Fig), indicating a remarkably early origin. This regulatory mechanism is functionally distinct from the autoregulation carried out by many RBPs that generate unproductive mRNA isoforms to control their own expression levels. In essence, the *MBNL* feedback loop seems to function as a developmental timer that generates a cytoplasmic pool of MBNL proteins only at an appropriate concentration of nuclear MBNL or, in a cell-specific context, under the control of co-regulating RBPs. In conclusion, our results in *C. elegans* demonstrate that expression of the shorter cytoplasmic MBL-1 isoforms is a crucial determinant for normal lifespan, neuromuscular function, and neuronal development, and point at an important role for MBNL proteins as regulators of miRNA expression that awaits further exploration.

## Materials and methods

### Ethics statement

We affirm that we have complied with all relevant ethical regulations for animal testing and research. Given that our experiments focused exclusively on *C. elegans* nematodes, no ethical approval was required for any of the presented work.

### *C. elegans* strains

Animals were maintained on NGM plates (peptone, agar and NaCl; Merck) at 20°C at all times. The N2 (Bristol) strain was used as the wild-type. The following strains were obtained from the *Caenorhabditis* Genetics Center (CGC): CZ10175 z*dIs5[Pmec-4::GFP + lin-15(+)]*, TY2685 *fox-1(y303)*, and VC446 *alg-1(gk214)*. The *mbl-1(tm1563)* strain was obtained from the National BioResource Project (NBRP). The following strains were custom generated through CRISPR/Cas9 genomic editing by SunyBiotech (Fuzhou, China): PHX4318 *mbl-1 (syb4318)*, referred to as *mbl-1 short*$^{(ex7-)}$; PHX4345 *mbl-1(syb4345)*, referred to as *mbl-1 long*$^{(ex7+)}$; and PHX5299 *mbl-1c(syb5299)*, the 3XFLAG::mCherry::MBL-1 strain. For a list of all strains used and created within this study, see S1 Appendix. For sequences of the edited genomic regions, see S2 Appendix. All strains were outcrossed at least five times with N2.

### Plasmid construction

To generate *Pmbl-1(1)::mbl-1-short::mCherry::unc-54 3′-UTR*, *Pmbl-1(2)::mbl-1-short:: mCherry::unc-54 3′-UTR*, *Pmbl-1(1)::mbl-1-long::mCherry::unc-54 3′-UTR* and *Pmbl-1(2)::mbl-1-long::mCherry::unc-54 3′-UTR*, we modified a plasmid expressing MBL-1 fused to mCherry

under the *unc-54* promoter [79]. For promoter 1, *Pmbl-1(1)*, primers 5′-TACGCATG-CAGGCCCTATATATTCCATCTCAAT-3′, containing a SphI site, and 5′-TACG-GATCCTCTGAAAAGTAGGAAAAAGATTGGC-3′, containing a BamHI site, were used. For *Pmbl-1(2)*, primers 5′-AACTGCAGGTGCAATGGGCTACTGATCTCC-3′ and 5′-CGGGATCCCATTCCGTCACTTGCAAAGAAC-3′ with PstI and BamHI sites, respectively, were used. Plasmids expressing short[(ex7-)] and long[(ex7+)] MBL-1 isoforms were derived from the resulting constructs through PCR using forward primers 5′-CAGCTA-CAAACTGCCGCCT-3′ (long) and 5′-GGAGCTGTACCAATGAAGCGAC-3′ (short), and reverse primer 5′-CTGATTCACTGCCGCTGCTGTATAAG-3′. For *Pmbl-1(1)::mbl-1-ex7 (+1)::mCherry::unc-54 3′-UTR* and *Pmbl-1(1)::mbl-1-ex7(-1)::GFP::unc-54 3′-UTR*, 2586 bp of a genomic region including sequences from *mbl-1* exon 6, 7 and 8 was amplified from N2 genomic DNA with primers 5′-GATCTAGCTAGCATGACTTCAAGAC-CCTTATACAGCAG-3′ and 5′-GATCTAGCTAGCATTGCTCCGTTCTTGTCGAGA-3′, both containing NheI sites. These fragments were then cloned into a modified pPD49.26 (Addgene) plasmid expressing mCherry and GFP, respectively, under the *Pmbl-1(1)* promoter with *unc-54* 3′-UTR [79]. Site-directed mutagenesis created frame shift-inducing single nucleotide insertions or deletions to ensure appropriate correspondence between splicing outcome and fluorescence expression.

## Transgenic animals

GAR153 was generated by co-injecting equal amounts of *Pmbl-1(1)::mbl-1-short::mCherry::unc-54 3′-UTR* and *Pmbl-1(2)::mbl-1-short::mCherry::unc-54 3′-UTR* plasmids into N2 animals. GAR154 was generated by co-injecting equal amounts of *Pmbl-1(1)::mbl-1-long::mCherry::unc-54 3′-UTR* and *Pmbl-1(2)::mbl-1-long::mCherry::unc-54 3′-UTR* plasmids into N2 animals. Plasmid concentrations for micro-injections were 15 ng/μl for the gene of interest, 5 ng/μl for a *myo-2::GFP* co-injection marker, and 65 ng/μl of empty pPD49.26 plasmid, respectively. The alternative splicing reporter strain GAR184 was generated by co-injection of *Pmbl-1(1)::mbl-1-ex7(+1)::mCherry::unc-54 3′-UTR* and *Pmbl-1(1)::mbl-1-ex7(-1)::GFP::unc-54 3′-UTR* plasmids into N2 animals at a concentration of 50 ng/μl for each construct. Extrachromosomal arrays were integrated into the *C. elegans* genome using gamma-ray irradiation with the OB29/4 unit (STS, Braunschweig, Germany). Integrated strains were outcrossed at least five times with N2.

## Soft touch assay

The assay was performed as in [80]. L4 stage animals were assayed while crawling on standard NGM plates seeded with OP50. Animals were gently stroked just behind the pharynx and anal region with an eyelash attached to a toothpick. Stopped movement or movement away from the eyelash was considered a positive response. Each animal was touched five times on the head and five times on the tail, and the number of positive responses was recorded for each animal. The assays were performed blindly regarding genotype, and for each strain, 60 animals were assayed grown on three different plates (20 animals per plate).

## Thrashing assay

L4 stage animals grown on NGM plates with OP50 were washed with M9 buffer, and then submerged with M9 buffer in a petri dish. Animals were left to acclimatize for one minute, and then recorded for 30 seconds using a Zeiss microscope (Axio Zoom v16) with camera attachment at 10x magnification at a rate of 80 frames per second. Recorded movies were analyzed using the Fiji-plugin wrMTrck [81, 82]. For each strain, three independent populations were analyzed grown on different plates.

## Lifespan assay

All lifespan assays were performed at 20˚C on standard NGM plates seeded with HT115. At the L4 larval stage, animals were transferred to plates containing 10 μM of 5-fluorouracil (Merck) to prevent progeny production. Animals with ruptured vulva or that crawled off the plate were censored. Animals were considered dead when there was no movement after repeated poking with a platinum wire. Lifespan analysis was performed using the OASIS application [83]. For all lifespan data, see S3 Appendix.

## RNAi

Animals were grown at 20˚C on standard NGM plates seeded with HT115. RNAi clones were taken from the Ahringer library. To prevent developmental defects associated with *alg-1* RNAi treatment, animals were transferred only at the L4 stage on plates supplemented with 5 mM isopropyl β-d-thiogalactoside (IPTG), ampicillin (100 μg/ml) and 10 μM of 5-fluorouracil to prevent progeny production. Control animals were fed bacteria expressing empty vector L4440 (EV).

## RT-PCR and qPCR

Animals were grown at 20˚C on standard NGM plates seeded with OP50 and collected at the L4 stage. After three washes with M9 buffer, RNA was isolated using the TRIzol reagent (Ambion). cDNA was made with the QuantiTect Reverse Transcription Kit (Qiagen), and standard RT-PCR was performed using the Phusion polymerase (Thermo Fisher) for 30 cycles in a Bio-Rad S1000 thermocycler. After gel electrophoresis, amplicons were analyzed on a 1% agarose gel. SYBR green I reagent (Roche) was used for qPCR reactions run in a Lightcycler 480 system (Roche). Results from qPCR analysis are derived from three biological repeats (each with three technical replicates) with RNA collected from animals grown on different plates. For a list of primers used for standard RT-PCR and qPCR, see S4 Appendix. qPCR data are summarized in S5 Appendix.

## Phylogenetic analysis

To identify MBNL orthologs, we performed a BLASTP [84] search on the National Center of Biotechnology Information website (ncbi.nlm.nih.gov) using the conserved sequence "WLQLEVCREFQRNKCTRPDTECKFAHP" from human MBNL1 as a query. We then performed a "KR motif" screen on MBNL orthologs from representative species within each clade, and evaluated the presence of this motif within alternative splicing isoforms using available data from the Ensembl.Metazoa (https://metazoa.ensembl.org/index.html) and the NCBI websites. KR motif encoding exons, *mbl-1* promoter regions and 3′-UTRs were aligned with MUSCLE [85] from the EMBL-EBI website and the phylogenetic tree for the TZF motifs was obtained using COBALT [86]. Conserved miRNA binding sites were found through the TargetScanWorm website. Sequence logos were created with WebLogo [87].

## Fluorescence microscopy

Animals were picked onto 3% agarose pads and immobilized with a drop of 50 mM levamisole (Merck). Images were taken using the Zeiss Axio Imager M2 fluorescence microscope, and relative fluorescence intensities and receptive field gap was measured using the ZEN 2.3 software (Zeiss). For experiments assaying the alternative splicing reporters, results were obtained from two independent populations. For experiments that measured receptive field gap, results were obtained from three independent populations (S5 Appendix).

### RNA-seq

Animals were synchronized by bleaching, plated as L1 larvae on NGM plates seeded with OP50, and collected at the L4 stage. Following RNA isolation with the TRIzol reagent (Ambion) with three biological replicates for each strain, the integrity of the RNA was assessed with the Agilent 2100 Bioanalyzer. PolyA-mRNA enriched library was prepared and sequenced at Novogene (Cambridge, United Kingdom) on the Illumina NovaSeq PE 150 platform. Sequencing yielded circa 60–75 million reads per sample. Reads were then mapped to the *C. elegans* reference (WBcel235 assembly) using the HISAT2 algorithm. The DESeq2 R package [88] was used for differential expression analysis, and for detection of differential alternative splicing, rMATS [89] was used. For small RNA-seq, libraries were prepared by ligation of adapters to the RNA ends and subsequent PCR enrichment of cDNA. After purification and size selection, libraries with insertions between 18 ~ 40 bp were ready for sequencing on Illumina NovaSeq SE50, yielding circa 9–16 million reads. Small RNA reads were mapped to the reference genome (WBcel235 assembly) using Bowtie [90], and the expression of known and unique miRNAs in each sample was statistically analyzed and normalized by TPM [91]. To calculate differential expression level, the DESeq2 R package was used. For library preparation of RIP samples, unenriched RNA samples were fragmented to 250 bp and reverse transcribed to cDNA with random primers at Novogene. Sequencing was performed on Illumina NovaSeq PE150, yielding circa 21–25 million reads. Reads were mapped on the reference transcriptome (WBcel235 assembly) with BWA [92]. Peak calling was done using MACS2 [93], and Diffbind [94] was used to call differentially represented peaks between input and immunoprecipitated samples. Results from three RIP-seq replicates were combined.

### RNA immunoprecipitation

Protocol was adapted from [95]. L4 stage 3XFLAG::mCherry::MBL-1 animals were collected and washed three times with ice-cold M9 buffer and then irradiated at 3000 mJ/cm$^2$ in a Stratalinker UV 1800 Crosslinker (Stratagene). Samples from triplicate biological repeats were washed with 1x RIP buffer (50 mM HEPES pH 7.4, 70 mM potassium acetate, 5 mM magnesium acetate, 0.05% NP-40 (Merck), and 10% glycerol), and supplemented with cOmplete proteinase inhibitor (Sigma-Aldrich) and RNAsin Plus RNAse inhibitor (Merck). Pellets were flash frozen with liquid nitrogen and ground with sterile mortar and pestle in cold room. Samples were then collected in 1x RIP buffer and placed on ice. ANTI-FLAG M2 magnetic beads resin (Merck) was washed twice with 1x TBS buffer (50 mM Tris-HCl, 150 mM NaCl, pH 7.4) and once with 1x RIP buffer. Cell lysates were added to the beads, and samples rotated for two hours at 4˚C. Beads were then washed five times in a high salt wash buffer (16 mM Tris, 500 mM NaCl). Elution by pH was performed at room temperature by incubating the beads for five minutes with 0.1 M glycine HCl (pH 3.0), and eluate neutralized with 0.5 M Tris (pH 7.4) and 1.5 M NaCl. RNA was isolated using the TRIzol reagent (Ambion) with acid-phenol:chloroform at pH 4.5.

### Western blot

Lysates from the RNA immunoprecipitation were resolved on 4–15% precast polyacrylamide gels (Bio-Rad). After protein transfer on nitrocellulose membrane, primary antibodies anti-flag (F1804, Merck) and anti-α-tubulin (T5168, Merck) were used in 1:2000 and 1:5000 dilutions, respectively. HRP-conjugated secondary goat anti-mouse antibody (ab97023, Abcam) was used in 1:2000 dilution, and chemiluminescent signal detected with enhanced chemiluminescent Western Blotting substrate (32106, Pierce) using the Fujifilm LAS-3000 imaging system.

## Statistical analysis

Statistical analysis (Student's *t*-tests, ANOVAs, Kruskal-Willis tests) for qPCR, fluorescence intensity analysis, thrashing assay, soft touch responsiveness, alternative splicing reporter analysis and receptive field gap measurement was performed using GraphPad Prism version 8.0.0 for Windows. A D'Agostino-Pearson test was used to test for normal distribution of the data, and Bartlett's test was performed to test for equal variance. For lifespan analysis, statistical significance was assessed through a log-rank test with the OASIS application. Cox proportional hazards regression analysis was performed at the http://datatab.net website.

## Supporting information

**S1 Fig. Evolutionary conservation of *mbl-1* promoter region 2 in nematodes.** Depiction of the *mbl-1* genomic region and DNA alignment of the genomic sequence surrounding exons 1 and 2 in the nematode lineage. Non-coding sequences are in lower case, coding sequences in upper case with potential start codons highlighted in yellow. Lower box depicts alignment of amino acid sequence encoded by exons 1 and 2, with the last amino acid encoded on exon 1 depicted in red. Asterisks indicate that the amino acid or nucleotide is the same for all sequences; a colon indicates that amino acids have similar chemical properties.
(TIF)

**S2 Fig. Evolutionary conservation of sequences in nematode *mbl-1* terminal exons and 3′-UTRs.** Depiction of the *mbl-1* terminal exons and 3′-UTRs, and DNA alignment in the nematode lineage. Non-coding sequences are in lower case, and coding sequences in upper case. Highlighted in yellow are respectively the 5′ splice site and 3′ splice site of the alternative splicing event. Stop codons are highlighted in blue, conserved miRNA binding sites in green (miR-79 and miR-58/80/81/82, respectively). Asterisks indicate that the nucleotide is the same for all sequences.
(TIF)

**S3 Fig. Evolutionary conservation of KR motif-associated alternative splicing in bilaterian MBNL proteins.** Phylogenetic representation of the presence of MBNL proteins within the animal kingdom, the double KR motif in two neighboring exons, and annotated alternative splicing associated with the KR motifs. Dots represent presence, hyphens absence, and question marks unidentified.
(TIF)

**S4 Fig. Evolutionary conservation of MBNL tandem zinc finger (TZF) motifs 1 and 2.** Protein alignment of TZF motifs 1 (top) and 2 (bottom). Amino acid sequence conservation > 85% is highlighted in yellow. Unrooted radiation tree depicts phylogenetic relationship between TZF motifs 1 (1) and 2 (2). The scale bar indicates 0.4 (40%) genetic variation for the length of the scale.
(TIF)

**S5 Fig. Alternative splicing reporter expression under promoter 1 versus promoter 2.** DIC and fluorescence images taken at the L4 stage of alternative splicing reporter animals for which GFP and mCherry genes were expressed under promoter 1 (panels I, III and V) or promoter 2 (II, IV and V), respectively. Arrows identify regions of differential expression; hn: unidentified sensory head neurons with dendrites on the dorsal site, sp: spermathecal, uv: unidentified cells in vulval region (putative uterine muscle cells), tn: unidentified tail neuron.
(TIF)

**S6 Fig. Fluorophore gene switching does not affect fluorescence ratios.** DIC and fluorescence images of alternative splicing reporter animals for which GFP and mCherry genes were switched (panels VII to X) compared to original splicing reporter (panels III to VI), taken at the early adult stage. Schematics showing the gene constructs used for the alternative splicing reporter strain. "D" and "v" indicate dorsal and ventral sides, respectively.
(TIF)

**S7 Fig. Fluorescence expression in posterior gut persists after 3′-UTR switching in the minigene constructs.** DIC and fluorescence images of alternative splicing reporter animals for which the *unc-54* 3′-UTR was replaced with the *let-848* 3′-UTR, taken at the L3 stage. Schematics showing the gene constructs used for the alternative splicing reporter strain. "D" and "v" indicate dorsal and ventral sides, respectively.
(TIF)

**S8 Fig. Evolutionary conservation of exon 7, MBL-1 and FOX-1 binding sites in the nematode *mbl-1* gene.** Multiple sequence alignment of the *mbl-1* genes from five representative species of the *Caenorhabditis* genus. Exon 7 in red, putative conserved MBL-1 binding sites (YGCY) in yellow, and putative conserved FOX-1 binding site (GCAUG) in blue. Asterisks indicate nucleotides with 100% sequence conservation.
(TIF)

**S9 Fig. FOX-1 does not regulate *mbl-1* AS in the VNC.** Quantification of relative mCherry fluorescence intensity in the VNC neurons from indicated animals expressing the fluorescence alternative splicing reporter minigenes at L4 stage. For each strain, a total population of 30 worms was analyzed, grown on two different plates (15 animals per plate). An unpaired two-sample Student's *t*-test was performed (ns: no significance). Error bars indicate SEM.
(TIF)

**S10 Fig. Transcript levels of *mbl-1* are not altered in *mbl-1* mutants.** Transcripts levels of *mbl-1* from L4 animals were measured by qPCR. Experiments were done with three biological repeats (each with three technical replicates) with RNA collected from animals grown on different plates. One-way ANOVA with Dunnett's multiple comparisons test was performed and no significant differences were found. Error bars indicate SEM.
(TIF)

**S11 Fig. Alternative splicing of *mbl-1* exon 7 is not affected by the absence of MBL-1 in the excretory canal cell.** Quantification of relative mCherry fluorescence intensity in the excretory canal cells from animals at the L4 stage expressing the fluorescence alternative splicing reporter minigenes. For each strain, a total population of 30 worms was analyzed, grown on two different plates (15 animals per plate). One-way ANOVA with Dunnett´s multiple comparison test was performed (*** $P < 0.001$; ns: no significance). Error bars indicate SEM.
(TIF)

**S12 Fig. mCherry::MBL-1 rescues splicing defects in *mbl-1(tm1563)*.** RT-PCR analysis of mCherry::MBL-1 splicing functionality at the L4 stage.
(TIF)

**S13 Fig. mCherry::MBL-1 is expressed in neurons in tail, head and VNC.** Representative DIC and fluorescence images of adult 3XFLAG::MBL-1::mCherry animals. Images represent tail (panels I and II), head (III and IV) and vulvar region (V and VI). Green signals are the

result of autofluorescence.
(TIF)

**S14 Fig. MBL-1 regulates alternative polyadenylation of the *pqn-52* gene.** Sashimi plots of the *pqn-52* gene from RNA-seq data derived from animals at the L4 stage showing regulation of alternative polyadenylation by MBL-1. Gene model with exons in blue boxes at the bottom. Numbers represent exon spanning reads.
(TIF)

**S1 Table. List of dysregulated splicing events in *mbl-1* mutants.**
(XLSX)

**S2 Table. Results from SRI analysis for *mbl-1 short*^(ex7-) and *mbl-1 long*^(ex7+) mutants.**
(XLSX)

**S3 Table. Differential gene expression analysis of RNA-seq from *mbl-1* mutants.**
(XLSX)

**S4 Table. Comparison of upregulated genes in *mbl-1 short*^(ex7-) mutants with transcriptional profile of mutants in the WormExp database.**
(XLSX)

**S5 Table. List of genes with transcripts pulled down together with MBL-1 by RIP-seq.**
(XLSX)

**S6 Table. Differentially expressed miRNAs in *mbl-1* mutants.**
(XLSX)

**S1 Appendix. List of all strains used and generated in this study.**
(DOCX)

**S2 Appendix. Sequences of CRISPR/Cas9-edited genomic regions.**
(DOCX)

**S3 Appendix. Results from lifespan assays.**
(DOCX)

**S4 Appendix. Oligonucleotide sequences used for RT-PCR splicing analysis and qPCR.**
(DOCX)

**S5 Appendix. Raw qPCR data and numerical values used to generate graphs.**
(XLSX)

## Acknowledgments

We are grateful to the CGC, funded by NIH Office of Research Infrastructure Programs (P40 OD010440), which provided some of the strains. We also thank the laboratories of Drs. Brendan Battersby and Olli Matilainen for sharing reagents, and Dr. Mikko Frilander for critical reading of the manuscript.

## Author Contributions

**Conceptualization:** Jens Verbeeren.

**Data curation:** Jens Verbeeren.

**Formal analysis:** Jens Verbeeren, Joana Teixeira.

**Funding acquisition:** Jens Verbeeren, Susana M. D. A. Garcia.

**Investigation:** Jens Verbeeren, Joana Teixeira.

**Methodology:** Jens Verbeeren, Susana M. D. A. Garcia.

**Project administration:** Jens Verbeeren, Susana M. D. A. Garcia.

**Resources:** Jens Verbeeren, Susana M. D. A. Garcia.

**Software:** Jens Verbeeren.

**Supervision:** Susana M. D. A. Garcia.

**Validation:** Jens Verbeeren.

**Visualization:** Jens Verbeeren.

**Writing – original draft:** Jens Verbeeren.

**Writing – review & editing:** Joana Teixeira, Susana M. D. A. Garcia.

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
