## [Decision Letter · Decision Letter 0]

21 Jul 2023

Dear Dr Garcia,

Thank you very much for submitting your Research Article entitled 'The Muscleblind-like protein MBL-1 regulates microRNA expression in Caenorhabditis elegans through an ancient autoregulatory mechanism' to PLOS Genetics.

The manuscript was fully evaluated at the editorial level and by independent peer reviewers. The reviewers appreciated the attention to an important problem, but raised substantial concerns about the current manuscript at multiple levels. Based on the reviews, we will not be able to accept this version of the manuscript, but we would be willing to review a much-revised version that fully addresses concerns raised by all three reviewers. We cannot, of course, promise publication at that time.

Should you decide to revise the manuscript for further consideration here, your revisions should address the specific points made by each reviewer. A revised manuscript should also include a complete strain/genotype list and access to any information deposited online.  We will require a detailed list of your responses to all of the review comments and a description of the changes you have made in the manuscript.

If you decide to revise the manuscript for further consideration at PLOS Genetics, please aim to resubmit within the next 60 days, unless it will take extra time to address the concerns of the reviewers, in which case we would appreciate an expected resubmission date by email to plosgenetics@plos.org.

We are sorry that we cannot be more positive about your manuscript at this stage. Please do not hesitate to contact us if you have any concerns or questions.

Yours sincerely,

Anne C. Hart

Academic Editor

PLOS Genetics

Gregory P. Copenhaver

Editor-in-Chief

PLOS Genetics

Reviewer's Responses to Questions

**Comments to the Authors:**

Reviewer #1: Review is uploaded as an attachment.

Reviewer #2: In this paper, Jens Verbeeren and Susana Garcia address the biological significance of two Muscleblind protein isoforms in C elegans. Thanks to the genetic amenability of the worm, they generate a number of interesting genetic tools: isoform-specific reporters, a bichromatic reporter construct, and worms that exclusively express one of the two isoforms. Combining gene expression and tissue analyses, and omics approaches (RNA-seq, small RNA-seq, and RIP-seq), they conclude that Mbl1-short and long have fundamentally different molecular and biological roles associated with their differential subcellular localizations. It is to be noted that few papers have previously approached the issue of the differential functions by Mbl isoforms, particularly in an in vivo context. The manuscript is very well written, it is easy to follow, and the conclusions are supported by evidence. I believe the manuscript is close to publishable but would like to know the authors' response to the following comments:

1.- Please clarify whether "replicated repeats" in different places in the manuscript refer to technical replicates (e.g., three RT-PCR from the same RNA extracts) or experimental replicates (RT-PCRs performed with RNA extracts from three independent biological materials).

2.- Line 144 indicates "under both promoter 1 and 2", but it is unclear along the manuscript which of the two promoters is shown every time or whether both promoters are included in the same construct or other possibilities. Please clarify

3.- In section 3 and other places, I´d like the authors to reason why they conclude Mbnl1 short is "required", instead of "sufficient", for normal lifespan. Because in their manipulation they remove the Mbl1 long and maintain the short isoform, it is indeed difficult to classify whether this is a loss or gain of function evidence. However, considering "required/necessary" as an interpretation of experiments where you remove a causal element and "sufficient" when you put back in the system that element, I think it´d be more appropriate to conclude that Mbl1-short is sufficient for normal lifespan (and that Mbl-long is not required, if authors want). 

4.- Upon CRISPR editing, authors must compare Mbl1 levels previous to the manipulation and after. This is to rule out potential overexpression effects associated to the manipulation and assign phenotypes exclusively to the loss of function of the missing isoform. This is, can the authors confirm no significant changes in protein expression after generating Mbl-1 short and long, and after other genetic manipulations that introduced reporters (e.g. line 331)? 

5.- The existence of two Muscleblind promoters was also previously reported for Drosophila and humans (e.g. https://journals.plos.org/plosone/article?id=10.1371/journal.pone.0093125), perhaps worth mentioning in the context of phylogenetic conservation.

6.- It is unclear whether additional Mbl-1 isoforms exist in C.elegans or if these two are the only ones. Please clarify and discuss.

Minor issues

1.- note the current name for MBNL: https://www.ncbi.nlm.nih.gov/gene/4154

2.- Myotonic dystrophy type 1 and 2 is far more common than with Roman numerals I and II

3.- Use "Student´s"

Reviewer #3: In this manuscript Verbeeren and Garcia investigate the RNA-binding protein MBL in C. elegans in particular with regards to its role in alternative splicing of itself as well as other genes. The authors report a developmental shift in isoform expression from a long (nuclear, earlier) version to a shorter (cytoplasmic, later) version of the protein. They then go on to test candidate genes and cellular sites for their role as co-regulators of mbl-1, and perform an RNAseq and lifespan analysis where they single out several genes they suggest indicate mbl-1 is a miRNA expression regulator.

I found the study to present several interesting and important results to the field even in the face of significant structural challenges. This is clearly a worthwhile study and I hope the authors will take the time to address the concerns/questions that arose during my reading of their work. I will next describe these criticism, which as I indicated do not detract from the importance or interest of the study.

1) Study Design: I must admit that I struggled to understand some of the study design. Some of the molecular manipulations were unclear to me. For example, the study assumes the existence of only two mbl-1 isoforms despite the literature evidence to the contrary. In figure 1 two promoters are shown for a long and a short isoform of mbl-1. It was not immediately obvious if the use of Promoter 2 always resulted in a long isoform including exon 7, and the use of Promoter 1 always resulted in a shorter isoform lacking it. The diagrams in the figures did not really help me understand this. Particularly in figure 1 where in B two promoters (1 and 2) are shown driving expression of the mCherry reporter. I spent a longtime looking at this diagram and reading the text and am still confused about the meaning. I personally think that a much cleaner experiment would have been to use a null mbl-1 mutant that removes all known mbl-1 isoforms (such as mbl-1(wy560) reported by Spilker et al.), and then introduce the mCherry tagged rescue constructs into that strain. This would have allowed the authors to make phenotypic assignments without worrying about alternative isoform expression. Furthermore, because the coding sequence for the longest mbl-1 isoform is still relatively short (<3kb), it would have been possible for the authors to investigate the features of the KB region in exon 7 responsible for nuclear localization. This could be done relatively easily using even an extrachromosomal array to drive the expression of a GFP-tagged MBL-1 isoform where site-directed mutagenesis was used to mutate the KR motif or other (control) exonic regions. That would have significantly improved the resolution power of the study. While discussing transgenic strains, the CRISPR/Cas9 strains that are reported to exclusively express either long or short isoforms are reported in the methods to have been outcrossed at least 5 times with N2 animals. Was the genotype of the resulting crosses confirmed through sequencing.

2) Context of the study: There is some discrepancy on the literature with regards to the actual number of mbl-1 isoforms. While the authors cite these studies, they make no mention of this discrepancy nor assess the impact on their results. For example, Wang et al (2007) reported two mbl-1 isoforms, but later work by others (i.e. Segawa et al., 2009, Spilker et al., 2012) reported six isoforms of mbl-1. This is further (and more recently) supported by the CenGen isoform analysis (with CENGEN actually providing evidence for eight isoforms, http://splicing.cengen.org/gene/WBGene00019347/). This is a rather important detail to leave out because there is no telling how these additional isoforms might impact the experiments performed or the results observed. One example that comes to mind involves the mutants used. While the mutation in tm1563 results in an in-frame deletion for longer isoforms, for the three shorter isoforms that start in exon-3 the mutation results in a frame-shift. This means that this mutation does not affect all available isoforms the same way.

3) Comparisons: At times the authors test and compare L3 larva, other times L4, yest other times Day-1 adults (e.g. behavioral assays). I think that the study could be improved by selecting and testing a single stage, or alternatively reporting their findings for all stages. If fluorescence microscopy limits comparison to L3 animals, then this stage should be compared throughout (e.g. behavioral tests, qPCR, etc). The present approach clouds interpretation of the results (at least those involving comparisons across life stages). In figure 1 we see mbl-1 expression in head and tail ganglia, ventral nerve chord, and spermatheca, but in the alternative splicing reporters is seen in ventral nerve chord but not in the head or tail ganglia. How do the authors account for this differential expression?

4) RNAi experiments: With regards to the inactivation of fox-1 through RNAi: was this done in a neuronally sensitized strain? It is known that many tissues in C. elegans are refractory to RNA interference and require the ectopic expression of a double stranded RNA channel to be successfully silenced. Because of this, silencing of fox-1 or any other gene in the absence of a sensitized background could only be expected to alter expression in tissues that are non-refractory to RNAi (e.g. glia and muscles) but not in neural tissues.

5) Behavioral assays: what is the estimated ISI for the repeated touch stimulation and how this compares to the results in the literature? Was ISI kept to 30 sec, 60 sec, etc.? What frame rate and magnification was used for the swimming movies. The phenotypes investigated were also vague or generic and do not provide opportunities to generate mechanistic insights with regards to the isoforms function. For example, just about any deleterious mutation or manipulation could be predicted to affect an animal’s lifespan or motility. Going after more specific phenotypes would have been more informative in this part of the study.

6) Longevity study: With regards to the study of longevity in C. elegans using FUDR, its use has recently been challenged by the finding that 5-fluorouracil itself has complex effects on lifespan which can cloud the interpretation of results (DOI: 10.1266/ggs.15-00064). It would therefore be recommended to validate the main results without using FUDR to control for any potential artifacts. Furthermore, was mRNA levels for the two isoforms quantified to control for dosage differences potentially driving longevity differences observed between the treatments?

7) Statistics: The authors used t-tests almost universally to compare all their data (except for the survival curves). However, t-tests require a very specific set of circumstances to be validly performed. T-tests must be done to compare two groups only (more groups require ANOVAs), and those two groups must satisfy two requirements: 1) normal distribution, and 2) equal variance (it is possible under certain conditions to use parametric tests in non-normally distributed data, but it is not clear if these conditions were met in this study). Throughout the study, it is clear that the authors used t-tests when they should have used ANOVAs, and there is to mention to performing tests of normality or equal variance. In some comparisons the variances are clearly different and would invalidate t-tests or ANOVAS (requiring ranked sum tests, or ANOVAS on ranks). Sample sizes are not usually provided although, I eventually figured (from the appendix Table S6) that most comparisons involved samples of just ten animals (Figure 1D is an exception and no raw data or N is provided for these data). One of the great advantages of working with C. elegans is their low cost and the great number of animals that can be brought to bear in experiments. It is therefore underwhelming that the study used only ten animals for most of their comparisons. Furthermore, from what I can infer these were animals that all were measured during a single assay? if all ten animals tested come from a single plate, there is the possibility that differences in cultivation (eg. temperature, bacterial or fungal infection, etc) could significantly alter one of the plates and therefore the results reported. For this reason, it is common to report the results of several assays performed over several days to minimize this type of error. I would therefore recommend that the authors justify the sample size they used in the context of power statistics (the beta value for their comparisons) or in the context of similar studies in the field. Although I will point out that most researchers would compare the means of many assays, rather than the mean of a single assays. As mentioned above, student t-tests are not the way to compare the swimming and touch assays, or any of the other comparisons involving multiple groups. If there are two or more manipulations (and there is one control group) the appropriate approach is to perform an ANOVA (if all groups are normally distributed and pass the equal variance test, or an ANOVA of ranks otherwise). Following the ANOVA, a Dunnett's test may be used for relevant treatments to be compared against a control group (N2s in this case).

8) Figures: The figures of the study were not up to publication standard. For example, there is not a single scale bar in any of the confocal images shown. There is also no consistency in orientation or description of orientation. Animals are presented head to the left or right in the same figure and often ventral and dorsal directions are left unresolved. Many composite plates do not have sub-indexes (eg. i, ii, iii). However sometimes the authors use non-standard quotations (‘, ‘’, ‘’’). Another inconsistency is with the choice of plots used. It is unclear what the justification for the variability in choice of error measurement reported throughout the study is. For example, in Figure 3C error bars represent 1.5 IQR but in Figure 3D and F error bars represent SEMs. In some figures data is presented as vertical bars with mean and SEM errors (Figure 1D), in other figures the authors use point plots and mean bars (Figure 2D E and G), other times they use box plots with IQR (Figure 3C), and yet other times they use violin plots (Figure 4C). The authors should stick to a consistent representation of similar data to avoid confusing the reader.

9) Discussion: In the context of DNA transcription and gene splicing I would not go as far as the authors to claim that the mechanism investigated displays “extreme” conservation. It might be conserved, but not any more conserved than other mechanisms associated with gene transcription and regulation. Much of the discussion was highly speculative and not directly supported by the data presented. Some of the interpretation was only supported in a tenuous manner. For example, the phenotypes investigated (e.g. lifespan and locomotion) are so broad that literally any insult to the organism would be predicted to affect them in these metrics. That does not mean that the insult is directly related or responsible for the phenotype. For example, one can predict that any illness might affect a person’s top swimming speed, be it a cold, Parkinson’s disease, arthritis, or a muscular sprain. This does not mean that a gene related to these diseases are also responsible for swimming in humans. Similarly, showing that sick worms swim slower or live less than healthy ones does not mean a gene is associated with longevity or the production of locomotion. That type of phenotypic assignment would require more narrowly defined parameters, and additional experimental manipulations.

**Have all data underlying the figures and results presented in the manuscript been provided?**

Reviewer #1: **No: **I could not find any reference to the RNA Seq data being deposited at e.g. GEO or SRA

Reviewer #2: Yes

Reviewer #3: **No: **Data for Figure 1D not available

PLOS authors have the option to publish the peer review history of their article (what does this mean?). If published, this will include your full peer review and any attached files.

Reviewer #1: **Yes: **Adam Norris

Reviewer #2: No

Reviewer #3: No

---

## [Decision Letter · Decision Letter 1]

17 Nov 2023

Dear Dr Garcia,

Thank you very much for submitting your Research Article entitled 'The Muscleblind-like protein MBL-1 regulates microRNA expression in Caenorhabditis elegans through an ancient autoregulatory mechanism' to PLOS Genetics.

The manuscript was fully evaluated at the editorial level and by independent peer reviewers. The reviewers appreciated the attention to an important topic but identified some concerns that I ask you address in a revised manuscript. If all concerns are addressed, we may be able to avoid another round of review.

We therefore ask you to modify the manuscript according to the review recommendations. Your revisions should address the specific points made by each reviewer. As Academic Editor, I believe that these can be addressed by textual revisions and by providing us with access to the GEO data associated with the manuscript. The GEO database Frequently Asked Questions page says "*use the Reviewer access link near the top of your Series (GSExxx) record to create a reviewer token which provides anonymous, read-only access to your private submissions*." 

As Academic Editor, I also ask that you: a) double-check that gene names are not capitalized (Example S11 Fig title, supplementary file column headers/data, and other places), b) clarify if integrated array strains were backcrossed,  c) correct integrated array nomenclature (*iceIs50 *not *iceis50*), and d) include strains mentioned in the Materials and Methods in S1 Appendix.

Yours sincerely,

Anne C. Hart

Academic Editor

PLOS Genetics

Gregory P. Copenhaver

Editor-in-Chief

PLOS Genetics

Reviewer's Responses to Questions

**Comments to the Authors:**

Reviewer #1: I maintain my enthusiasm about the interesting observations presented in this manuscript, and my questions have mostly been addressed. A few final points remain-

Very important: the authors say the GEO submission is underway, but until it is publicly available, the current data availability statement of “All relevant data are available from within the manuscript as well as the supplemental information file” is not accurate. I do not know PLoS’ policy on this issue, the norm is that reviewers are permitted to view the data before recommending to accept.

Other points:

1. I am glad to see the new data on evolutionary conservation of the alternative exon. I’d still personally prefer the use of “ancient” to be changed to something more objective such as “evolutionarily conserved.”

2. The authors mention in the response to reviewers that it remains unknown which isoform of mbl-1 is predominant in touch neurons, but this is not mentioned in the text, as far as I can tell. It would be useful to add to the text in order to appropriately caveat the interpretation of the neurite outgrowth assays.

3. I am glad to hear the authors’ hypotheses in the response to reviewers about the varying degrees of lifespan deficits in mbl-1 mutants, but the manuscript text itself would also benefit from clarification. The text cites the literature that “…MBL-1 severely reduces lifespan,” but the cited references [37, 46] demonstrate mild lifespan reduction (OP50) or more severe reduction (HT115). As such a modification of the text to “…MBL-1 severely reduces lifespan when grown on HT115 bacteria” or simply “…MBL-1 reduces lifespan” seem more appropriate.

Reviewer #2: I think the authors did a great job addresing the different comments and recommendations I made and have no further comments.

Reviewer #3: This is a resubmission of the manuscript by Verbeeren and colleagues on the role of muscleblind protein regulating microRNA expression in C. elegans.

In this latest version of this manuscript, the authors have made an (honestly) impressive effort to address the reviewers' comments. This included a plethora or changes including to the introduction, methods, results, and discussion. Basically, every aspect of this manuscript has improved.

While not every criticism resulted in a change (most did), all were addressed and the remaining criticisms are considered (by this reviewer) as a difference of opinion that do not preclude accepting the manuscript. I think, and hope the authors agree, this is a vastly improved effort.

**Have all data underlying the figures and results presented in the manuscript been provided?**

Reviewer #1: **No: **

Reviewer #2: Yes

Reviewer #3: Yes

PLOS authors have the option to publish the peer review history of their article (what does this mean?). If published, this will include your full peer review and any attached files.

Reviewer #1: **Yes: **Adam Norris

Reviewer #2: No

Reviewer #3: No

---

## [Editor Report · Decision Letter 2]

12 Dec 2023

Dear Dr Garcia,

We are pleased to inform you that your manuscript entitled "The Muscleblind-like protein MBL-1 regulates microRNA expression in Caenorhabditis elegans through an evolutionarily conserved autoregulatory mechanism" has been editorially accepted for publication in PLOS Genetics. Congratulations!

Yours sincerely,

Anne C. Hart

Academic Editor

PLOS Genetics

Gregory P. Copenhaver

Editor-in-Chief

PLOS Genetics

Comments from the reviewers (if applicable):

**Data Deposition**

http://datadryad.org/submit?journalID=pgenetics&manu=PGENETICS-D-23-00671R2

**Press Queries**

---

## [Editor Report · Acceptance letter]

19 Dec 2023

PGENETICS-D-23-00671R2 

The Muscleblind-like protein MBL-1 regulates microRNA expression in Caenorhabditis elegans through an evolutionarily conserved autoregulatory mechanism 

Dear Dr Garcia, 

We are pleased to inform you that your manuscript entitled "The Muscleblind-like protein MBL-1 regulates microRNA expression in Caenorhabditis elegans through an evolutionarily conserved autoregulatory mechanism" has been formally accepted for publication in PLOS Genetics! Your manuscript is now with our production department and you will be notified of the publication date in due course.

With kind regards,

Anita Estes

PLOS Genetics

On behalf of:
